# Protein Design with Dynamic Protein Vocabulary

**Nuowei Liu**[1]*          **Jiahao Kuang**[1]*          **Yanting Liu**[1]

**Tao Ji**[2]          **Changzhi Sun**[3]          **Man Lan**[1]          **Yuanbin Wu**[1]

[1] School of Computer Science and Technology, East China Normal University
[2] English Department, College of Foreign Languages and Literatures, Fudan University
[3] Institute of Artificial Intelligence (TeleAI), China Telecom
{nwliu@stu, jhkuang@stu, ybwu@cs}.ecnu.edu.cn, taoji@fudan.edu.cn

## Abstract

Protein design is a fundamental challenge in biotechnology, aiming to design novel sequences with specific functions within the vast space of possible proteins. Recent advances in deep generative models have enabled function-based protein design from textual descriptions, yet struggle with structural plausibility. Inspired by classical protein design methods that leverage natural protein structures, we explore whether incorporating fragments from natural proteins can enhance foldability in generative models. Our empirical results show that even random incorporation of fragments improves foldability. Building on this insight, we introduce PRODVA, a novel protein design approach that integrates a text encoder for functional descriptions, a protein language model for designing proteins, and a fragment encoder to dynamically retrieve protein fragments based on textual functional descriptions. Experimental results demonstrate that our approach effectively designs protein sequences that are both functionally aligned and structurally plausible. Compared to state-of-the-art models, PRODVA achieves comparable function alignment using less than 0.04% of the training data, while designing significantly more well-folded proteins, with the proportion of proteins having pLDDT above 70 increasing by 7.38% and those with PAE below 10 increasing by 9.62%. [1]

## 1 Introduction

The natural world presents a remarkably intricate landscape of proteins that have evolved over billions of years to perform diverse biochemical functions, such as catalyzing reactions and binding specific molecules [1, 2, 3]. These natural proteins, found across all forms of life, represent only a small fraction of the vast sequence space of possible proteins. Designing novel proteins that exhibit user-specified functions remains a longstanding challenge in biotechnology.

Recent deep generative models have demonstrated promising potential in function-based protein design. They accept texts as input, either in the form of simple control tags (e.g., ProGen [4], ESM3 [5]) or complex function descriptions containing rich and nuanced information (e.g., ProteinDT [6], PAAG [7], Pinal [8]), and generate new protein sequences. One challenge (and a main shortcoming) of current models is that, besides satisfying the requirements of input texts, the designed protein should be able to fold into stable three-dimensional structures.

---

* Equal Contribution.
✉ Corresponding authors are Yuanbin Wu, Tao Ji, Changzhi Sun and Man Lan.
[1] Datasets and codes are publicly available at https://github.com/sornkL/ProDVa.

39th Conference on Neural Information Processing Systems (NeurIPS 2025).

To improve foldability, we notice that, though being expert-intensive and time-consuming, classical methods for protein design usually take inspiration from natural structures. For example, rational design [9, 10] leverages physical principles applied to known structures, and directed evolution [11, 12] relies on iterative cycles of mutation based on known structures. Therefore, for generative protein design models, a natural question is *whether well-folded novel proteins with user-specified functions can be directly assembled by utilizing fragments of natural proteins (e.g., motifs, functional sites, etc.) and their extensive functional annotations.*

Before jumping into learned generative models, we first consider two random generative models. Figure 1 presents a primary empirical analysis, illustrating that even the random incorporation of fragments into amino acid sequences can expand the landscape of generated proteins and enhance their foldability.

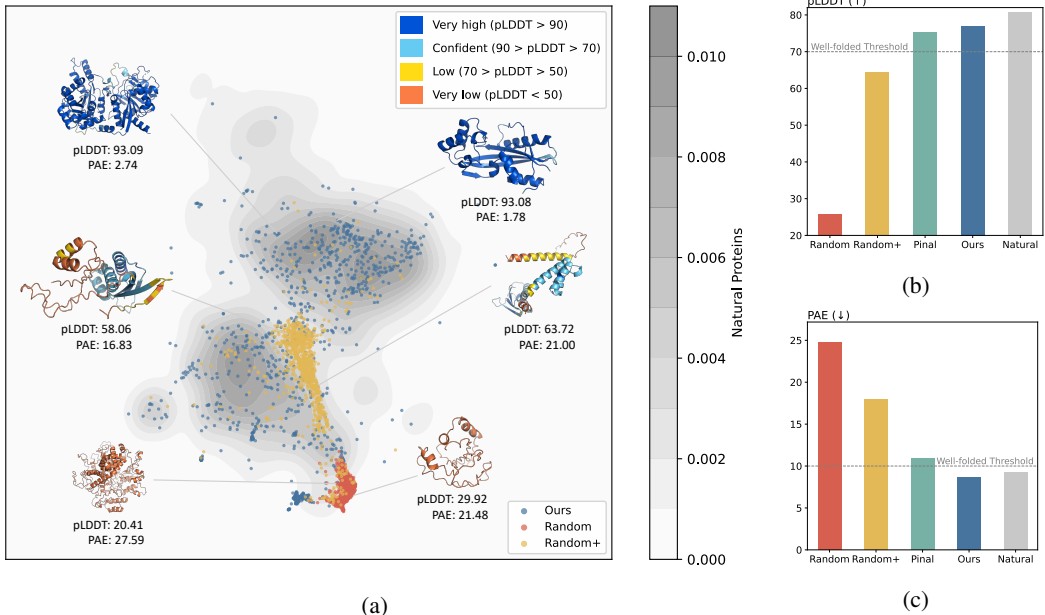

Figure 1: **(a)** Visualization of proteins designed by our method, embedded using ESM C [5] and projected with UMAP [13]. `Random` refers to proteins generated by randomly selecting amino acids according to their empirical distribution in SwissProt [14]. `Random+` refers to proteins generated by selecting amino acids and incorporating fragments. Natural proteins randomly sampled from SwissProt (in gray) form a broad distribution, representing the diverse landscape of natural proteins. Proteins generated by `Random` sampling from all possible sequences (in red scatter points) cluster tightly at the periphery of the natural protein distribution, suggesting that random proteins are less diverse. Proteins generated by `Random+` (in yellow scatter points) exhibit a much more diverse distribution than those generated by `Random`. **(b)** Performance on pLDDT (↑). Our method improves pLDDT by 12% over `Random+`, exceeding the well-folded threshold. **(c)** Performance on PAE (↓). Our method reduces PAE by 9% compared to `Random+` and is the only model to surpass the well-folded threshold. Notably, it outperforms the state-of-the-art baseline model Pinal in both metrics.

To further improve the random generative models, inspired by research on dynamic vocabulary [15, 16, 17], we propose a novel approach, PRODVA. Our method integrates a text language model for encoding functional descriptions and annotations, a protein language model backbone for generating *de novo* protein sequences, and a fragment encoder for learning protein fragment representations. Similar to other GPT-style models, PRODVA can be trained in a self-supervised manner, using protein sequences split into sets of tokens (amino acids) and fragments. During inference, protein fragments are dynamically retrieved as candidates according to various textual functions provided as inputs. In Figure 1, compared to the two random generative models, `Random` and `Random+`, our method (in blue scatter points) effectively spans the landscape of natural proteins and demonstrates a robust ability to design well-folded proteins. Furthermore, in comparison to the state-of-the-art model Pinal, PRODVA achieves competitive alignment with textual descriptions (a

marginal gap of 0.1%) using less than 0.04% of the training data, and significantly outperforms in foldability, generating 7.38% and 9.62% more structurally plausible proteins with pLDDT > 70 and PAE < 10, respectively.

## 2   Background

Proteins are macromolecules composed of linear chains of amino acids, with 20 standard amino acids serving as their fundamental building blocks. Continuous subsequences $S = \{s_1, s_2, \cdots, s_m\}$ within an amino acid sequence $P$, often referred to as fragments or segments, play crucial roles in determining the structure and function of proteins. In Figure 2, we utilize InterPro [18], an integrated database and diagnostic tool, to identify these fragments within proteins. These fragments can be classified into 8 categories: Domain, Family, Homologous Superfamily, Repeat, Conserved Site, Active Site, Binding Site, and Post-Translational Modification (PTM). Details of fragment types are discussed in Appendix B. The fragments, along with their corresponding types and descriptions, are referred to as functional annotations, denoted by $\mathcal{A}_{\text{type}}(S)$ and $\mathcal{A}_{\text{desc}}(S)$.

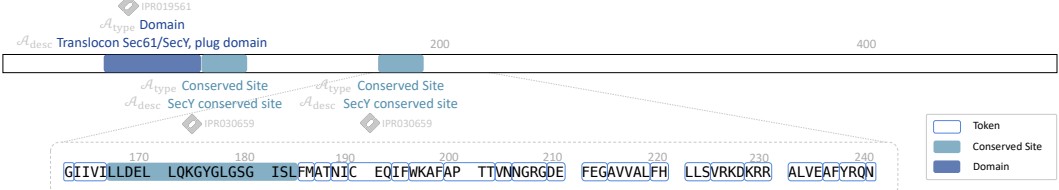

Figure 2: An example of Q96TW8, illustrating how an amino acid sequence is divided into sets of tokens and fragments. Note that when the BPE tokenizer [19] is used, a single token may represent multiple amino acids.

The function-based protein design problem is to generate a novel protein $P$ given the textual function description $t$. The problem can be formalized as a conditional generation task, denoted as $p(P \mid t)$. Our goal is to assemble a novel protein using the 20 standard amino acids $A = \{a_1, a_2, \cdots, a_{20}\}$ and a set of protein fragments $S$.

$$p(P \mid t) = p((x_1, x_2, \cdots, x_k) \mid t, \forall i, x_i \in A \cup S)$$

## 3   Method

### 3.1   Model Architecture

As illustrated in Figure 3, PRODVA consists of three components: (1) a Text Language Model (TextLM), (2) a Protein Language Model (PLM), and (3) a Fragment Encoder (FE).

**Text Language Model**   encodes the input textual function description $t$. To align the text representations $\mathbf{Z}_t = \texttt{TextLM}(t)$ with the embedding space of protein tokens, a projection layer is employed to transform $\mathbf{Z}_t$ into protein tokens $\mathbf{H}_t$.

**Protein Language Model**   serves as a decoder-only architecture backbone, capable of autoregressively designing *de novo* protein sequences. The PLM employs a fixed static vocabulary $V_{\text{tokens}}$ with a tokenizer that divides the protein sequences into discrete tokens. We denote the token embedding matrix and the language modeling head matrix of the PLM as $\mathbf{W}_{\text{tokens, in}} \in \mathbb{R}^{|V_{\text{tokens}}| \times d_{\text{PLM}}}$ and $\mathbf{W}_{\text{tokens, out}} \in \mathbb{R}^{d_{\text{PLM}} \times |V_{\text{tokens}}|}$, respectively, where $d_{\text{PLM}}$ is the hidden dimension of the PLM.

**Fragment Encoder**   Given a set of fragment candidates $S = \{s_1, s_2, \cdots, s_m\}$, where $m$ denotes the number of distinct fragments, the fragment representations can be trivially obtained using the Fragment Encoder, denoted as $\texttt{FE}(s_1, s_2, \cdots, s_m)$. By embedding the fragment representations through a projection layer, we obtain the fragment embedding matrix $\mathbf{W}_{\text{fragments}} \in \mathbb{R}^{m \times d_{\text{PLM}}}$. Note that the fragment representations are mapped to the same embedding space as the PLM.

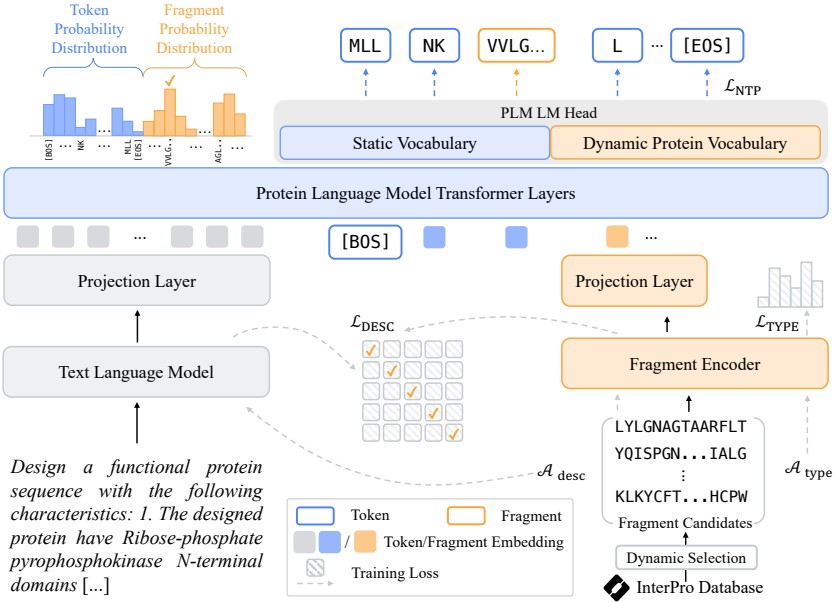

Figure 3: Overview of our model architecture.

## 3.2 Training

During training, given a training set $D' = \{(t_i, P_i)\}_{i=1}^N$ consisting of $N$ text-protein pairs, we first obtain all functional annotations using InterPro. For each minibatch $M = \{(t_i, P_i, \mathcal{A}_{\text{type}}(S_i), \mathcal{A}_{\text{desc}}(S_i))\}_{i=1}^B$, where $B$ is the inner batch size and each protein $P_i$ contains a variable number of segments $S_i$, the corresponding fragment types and descriptions are included.

**Learning Next Token/Fragment Prediction** To generate novel amino acid sequences, we adopt the standard language modeling objective of predicting the next token. There are two key differences.

First, the embedded input functional description $\mathbf{H}_t$ is incorporated as a prefix, serving as a conditioning context. Therefore, the preceding context, consisting of the functional description $t$ and the previous $i$ tokens or fragments, denoted as $\mathbf{H}_{\text{pre}}$, can be represented as

$$\mathbf{W}_{\text{in}} = \text{Concat}(\mathbf{W}_{\text{tokens, in}}, \mathbf{W}_{\text{fragments}})$$
$$\mathbf{H}_{\text{pre}} = \text{Concat}(\mathbf{H}_t, \mathbf{W}_{\text{in}}[x_{<i}]) \tag{1}$$

where $x_{<i}$ denotes the previous $i$ tokens or fragments.

Second, the prediction of the next token or fragment is determined by the joint distribution of the token and fragment probability distributions. Given the preceding context $\mathbf{H}_{\text{pre}}$ defined in Equation 1, the prediction of the $i$-th token or fragment can be computed as follows:

$$\mathbf{W}_{\text{out}} = \text{Concat}(\mathbf{W}_{\text{tokens, out}}, \mathbf{W}_{\text{fragments}}^{\text{T}})$$
$$p(x_i = k | \mathbf{H}_{\text{pre}}) = \frac{\exp(\mathbf{H}_{\text{pre}} \mathbf{W}_{\text{out}}^{(k)})}{\sum_{k' \in V_{\text{tokens}} \cup S} \exp(\mathbf{H}_{\text{pre}} \mathbf{W}_{\text{out}}^{(k')})} \tag{2}$$

Therefore, the learning objective of predicting the next token or fragment is to minimize the negative log-likelihood of Equation 2 as the following:

$$\mathcal{L}_{\text{NTP}} = -\frac{1}{n} \log \sum_{i=1}^{n} p(x_i \mid \mathbf{H}_{\text{pre}}) \tag{3}$$

**Learning Functional Annotations** To fully leverage the fragments and their corresponding functional annotations, $\mathcal{A}_{\text{type}}(\cdot)$ and $\mathcal{A}_{\text{desc}}(\cdot)$, we propose a type loss $\mathcal{L}_{\text{TYPE}}$, and a description loss $\mathcal{L}_{\text{DESC}}$.

The type loss is designed to capture the intrinsic relationships between fragments and their corresponding types by classifying the type. To this end, we add a MLP as the classification head, consisting of a linear layer with dropout [20] to the Fragment Encoder. Note that the fragment types may be unbalanced. Therefore, we assign a weight $w$ to each type across the entire training set $D$ instead of within each minibatch. The weighted type loss $\mathcal{L}_{\text{TYPE}}$ for the minibatch $M$ can be defined as follows:

$$\mathcal{L}_{\text{TYPE}} = -\frac{1}{\sum\limits_{i=1}^{B} |S_i|} \sum_{i=1}^{B} \sum_{j=1}^{|S_i|} w_{\mathcal{A}_{\text{type}}(S_i)_j} \log \left( \frac{\exp\left(\mathbf{q}_i^{(j,\mathcal{A}_{\text{type}}(S_i)_j)}\right)}{\sum\limits_c \exp\left(\mathbf{q}_i^{(j,c)}\right)} \right) \tag{4}$$

where $\mathbf{q}_i = \texttt{MLP}(\texttt{FE}(S_i))$ denotes the logits computed by the classification head and the Fragment Encoder.

Additionally, we define the description loss using the InfoNCE loss [21]. Similar to the type loss, we introduce an additional MLP as the description projection layer to the Text Language Model. We use the fragment representations $\mathbf{u}_i = \texttt{FE}(S_i)$ and their projected description representations $\mathbf{v}_i = \texttt{MLP}(\texttt{TextLM}(\mathcal{A}_{\text{desc}}(S_i)))$ as positive samples. The description loss aims to learn the relationships between fragments and their descriptions. Therefore, we select the remaining fragment-description pairs in a minibatch $M$ as negative samples. The description loss $\mathcal{L}_{\text{DESC}}$ is defined as follows:

$$\mathcal{L}_{\text{DESC}} = -\frac{1}{\sum\limits_{i=1}^{B} |S_i|} \sum_{i=1}^{B} \sum_{j=1}^{|S_i|} \log \frac{\exp(\text{sim}(\mathbf{u}_{ij}, \mathbf{v}_{ij})/\tau)}{\sum\limits_{k=1}^{B} \sum\limits_{l=1}^{|S_k|} \exp(\text{sim}(\mathbf{u}_{ij}, \mathbf{v}_{kl})/\tau)} \tag{5}$$

where $\tau$ is the temperature and $\text{sim}(u,v) = \frac{u^\top v}{\|u\|\|v\|}$ denotes the cosine similarity, which measures the similarity between the fragment representations and the description representations.

Note that the two additional MLP modules (i.e., the classification head and the description projection layer) are introduced during training to facilitate effective learning of the functional annotations and are discarded during inference. The final training objective is defined as

$$\mathcal{L} = \mathcal{L}_{\text{NTP}} + \alpha \mathcal{L}_{\text{TYPE}} + \beta \mathcal{L}_{\text{DESC}} \tag{6}$$

where $\alpha$ and $\beta$ are two significant hyperparameters used to adjust the weights of the loss function.

### 3.3 Inference

Compared to the training paradigm, the strategy for constructing fragment candidates during inference differs. Assume a set of supporting documents $D' = \{(t_i, P_i)\}_{i=1}^{N}$, consisting of $N$ text-protein pairs similar to those in the training set. Given a textual function description $t$ as input, the top $K$ most relevant descriptions are retrieved from the supporting documents based on the similarity of their description embeddings. InterPro is then used to identify the fragments of the $K$ protein sequences, resulting in the fragment candidates $\{S_i\}_{i=1}^{K}$ for inference. Additionally, we employ a top-k sampling decoding strategy to further enhance the diversity of the designed novel proteins.

## 4 Experiments

### 4.1 Experimental Setups

**Implementation Details**   The Text Language Model is initialized with GPT-2 [22] and its pretrained weights. Both the Protein Language Model and the Fragment Encoder are initialized with ProtGPT2 [23], which is a GPT-2 based PLM capable of generating *de novo* protein sequences under an unconditional generation setting. We employ PubMedBERT [24] as the embedding model to retrieve the top $K$ most similar descriptions using the txtai framework, which is supported by the Faiss [25] backend. Unless otherwise specified, the supporting documents used in our experiments are identical to the training sets to ensure a fair comparison, and we select $K = 16$ during inference. The results are averaged across three runs with different random seeds. Details of the training implementation are provided in Appendix C.1. An ablation study to investigate the effect of loss weighting is presented in Appendix E.4.

**Baselines** We compare PRODVA with the following function-based protein design methods. ProteinDT [6] is a multimodal framework consisting of a facilitator that generates protein representations and a decoder that creates the protein sequences from the representations. Pinal [8] first generates protein structures based on textual descriptions and then designs protein sequences from the structures.[2] PAAG [7] integrates the textual annotations and designs the proteins conditioned on flexible combinations of domain annotations. ESM3 [5] is a multimodal generative language model that generates proteins based on multi-hot function keywords. Chroma [26] is a diffusion model for protein generation that leverages the ProCap model for natural language understanding, which connects a GNN backbone with an autoregressive language model. Additionally, we implement three random baselines. `Random (U)` refers to the random selection of amino acids with a uniform distribution. `Random (E)` involves random selection based on the empirical amino acid distribution observed in SwissProt. `Random+ (E)` selects amino acids according to the empirical distribution with a 90% probability, while incorporating random fragments with a 10% probability.

**Metrics** To comprehensively evaluate our approach, we consider the following metrics from four perspectives. Detailed definitions and implementations are provided in Appendix D. **Sequence Plausibility.** For rapid validation of designed protein sequences, we employ Perplexity (PPL) and Repetitiveness (Rep) [27] to evaluate sequence-level plausibility. **Foldability.** Following [8], we utilize ESMFold [28] to predict the 3D structures of designed proteins and compute the predicted Local Distance Difference Test (pLDDT) [29] as a per-residue measure of local confidence (0-100, higher is better). Additionally, we also compute the Predicted Aligned Error (PAE) [29], which measures the confidence in the relative position of two residues within the predicted structure (lower is better). Consistent with previous researches [29, 30, 31, 8], we adopt pLDDT above 70 and PAE below 10 to define well-folded predicted structures. **Language Alignment.** To mitigate the high cost of wet-lab experiments, we employ oracle model-based metrics to evaluate the alignment between proteins and functions. Following [8], we use ProTrek [32], which evaluates alignment using cosine similarity, referred to as the ProTrek Score, within a joint embedding space. We also use EvoLlama [33], fine-tuned on our downstream datasets, computing the EvoLlama Score as the cosine similarity between textual descriptions and predicted functions. Higher scores from both models indicate better alignment with the specified function. Additionally, we employ two retrieval-based metrics. The first is Keyword Recovery [5], which measures the recovery of function keywords in the designed proteins using InterProScan [34]. And the second is Retrieval Accuracy [6], which assesses how well the designed proteins aligns with their functional descriptions by evaluating whether their representations exhibit the highest similarity among all candidate pairs. **Sequence Diversity.** MMseqs2 [35] is used to compute sequence-level similarity between each pair of proteins within a batch, where all sequences are generated based on the same textual description. A higher score indicates a greater ability to design diverse protein sequences.

## 4.2 Designing Proteins from Function Keywords

**Datasets** To ensure a fair comparison with ESM3, we use the same test set employed by ESM3. This test set is derived from the Continuous Automated Model EvaluatiOn (CAMEO) targets released between May 1, 2020 and August 1, 2023 [36]. Proteins lacking function keywords or containing only Family or Homologous SuperFamily keywords are excluded, resulting in a final test set comprising 507 proteins, referred to as the CAMEO subset. For training and validation, we construct the dataset by annotating the function keywords of proteins in SwissProt using InterPro, resulting in approximately 412K proteins. We randomly sample 5% of this dataset as the validation set, with the remaining used for training. Details are included in Appendix C.2.

**Results** Pinal is a state-of-the-art baseline model capable of designing well-folded, user-specified amino acid sequences. As shown in Table 1, Pinal consistently outperforms other baselines across most evaluation metrics. We highlight our key findings as follows:

**(1) Under the same training data setting, PRODVA consistently surpasses both ProteinDT and PAAG.** ProteinDT and PAAG show inferior performance even after fine-tuning with additional data consisting of function keywords. In contrast, PRODVA, trained on the same dataset, consistently surpasses these two models, achieving improvements of 34.35% in pLDDT and 17.51% in PAE.

---

[2] Pinal was trained on 1.76B pairs from SwissProt and UniProtKB, and has likely encountered most proteins and annotations in various forms. Thus, we do not distinguish baselines by zero-shot settings.

Table 1: Performance on the CAMEO subset (**Best**, Second Best). "Pairs" refers to the text-protein pairs used during training and "Params" denotes the model's parameter size. Violin plots are provided in Appendix E.6. † indicates that the training scripts are not publicly available.

| Models | #Pairs #Params | Sequence Plausibility | | Foldability | | | | Language Alignment (in %) | | | Sequence Diversity (↑) |
|---|---|---|---|---|---|---|---|---|---|---|---|
| | | PPL (↓) | Rep (↓) | pLDDT (↑) | % > 70 (↑) | PAE (↓) | % < 10 (↑) | ProTrek Score (↑) | Keyword Recovery (↑) | Retrieval Accuracy (↑) | |
| Natural | - | 467.64 | 0.02 | 81.21 | 90.53 | 7.08 | 82.05 | 21.09 | 100.00 | 70.81 | - |
| Random (U) | - | 2471.95 | 0.01 | 24.38 | 0.00 | 23.81 | 0.13 | 7.50 | 0.00 | 6.05 | 97.46 |
| Random (E) | - | 3046.64 | 0.01 | 27.46 | 0.00 | 23.70 | 0.00 | 6.59 | 0.00 | 5.13 | 99.78 |
| Random+ (E) | - | 966.24 | 0.01 | 62.38 | 32.65 | 17.23 | 9.28 | 3.29 | 0.00 | 5.79 | 98.97 |
| ProteinDT | 541K/729M | 1405.70 | 0.11 | 38.70 | 0.20 | 26.25 | 0.00 | 3.89 | 0.05 | 7.43 | 99.72 |
| ProteinDT$_{FT}$ | 392K/729M | 1860.43 | 0.04 | 38.66 | 1.04 | 23.90 | 0.42 | 6.28 | 1.08 | 16.57 | 99.32 |
| Pinal† | 1.76B/2B | 584.22 | 0.15 | 66.50 | 47.21 | 14.57 | 33.53 | **14.57** | **30.46** | **51.68** | 82.72 |
| PAAG | 130K/1.3B | 2571.40 | 0.02 | 33.14 | 0.00 | 23.31 | 0.00 | 5.21 | 0.23 | 7.10 | 99.02 |
| PAAG$_{FT}$ | 392K/1.3B | 2004.01 | 0.04 | 41.53 | 1.12 | 24.34 | 0.46 | 3.46 | 0.01 | 7.82 | **99.87** |
| Chroma† | 45K/334M | 1322.37 | 0.03 | 61.66 | 28.96 | 13.01 | 39.03 | 2.97 | 0.11 | 6.57 | 97.21 |
| ESM3† | 539M/1.4B | **279.78** | 0.33 | 59.79 | 31.49 | 17.40 | 21.37 | 3.76 | 5.49 | 11.97 | 96.77 |
| PRODVA | 392K/1.8B | 656.04 | **0.01** | **75.88** | **77.00** | **6.39** | **83.88** | 14.43 | 30.34 | 44.77 | 98.58 |

Furthermore, our approach surpasses ProteinDT by an average of 8.15% on the ProTrek Score and approximately 30% on Keyword Recovery, demonstrating the effectiveness of our proposed learning methods in low-resources settings.

**(2) PRODVA remains within a reasonable PPL range and demonstrates the capability to design well-folded proteins.** Although ESM3 achieves the lowest PPL, the sequences it generates display repetitive patterns, leading to relatively lower pLDDT scores and higher PAE values. Compared to PRODVA, the number of well-folded proteins is lower by 45.51% and 66.48% in terms of pLDDT and PAE, respectively.

**(3) PRODVA uses only 0.02% of the text-protein pairs used to train Pinal, yet achieves competitive performance.** Compared to PRODVA, Pinal shows marginal improvements, with increase of 0.14% and 0.12% on ProTrek Score and Keyword Recovery, respectively. Moreover, our method demonstrates significantly superior performance in foldability, outperforming Pinal by 9.38% in pLDDT and 8.18% in PAE, highlighting the effectiveness in designing structurally plausible proteins.

## 4.3 Designing Proteins from Textual Descriptions

**Datasets** Unlike ESM3, which accepts function keywords but lacks the understanding of natural language, PRODVA accepts arbitrary textual descriptions as input. Therefore, we utilize the Mol-Instructions [37] protein design-related instructions as the test set. The Mol-Instructions dataset comprises approximately 200K instruction-protein pairs, with each instruction generated from 20 UniProtKB-derived features [38] that capture both functional (e.g., catalytic activities, Gene Ontology annotations [39]) and structural properties (e.g., helix, turn). These structured annotations are converted into instructions using diverse templates enriched by LLMs. We combine the dataset with text-protein pairs derived from SwissProt, resulting in a total of 712K text-protein pairs used as the training set. Additionally, we allocate 10% of the Mol-Instructions protein design-related training set for validation. Details are included in Appendix C.2.

**Results** Table 2 presents a comparison between PRODVA and other baseline models on Mol-Instructions. Our key observations are highlighted as follows:

**(1) Most baseline models struggle to design proteins that are both well-folded and well-aligned, whereas PRODVA successfully accomplishes this.** Random+ (E) serves as a strong baseline, demonstrating that incorporating even random fragments can substantially improve the structural plausibility of proteins. Among the other baselines, only Pinal and Chroma are capable of designing well-folded proteins when compared to randomly generated sequences. However, the sequences generated by Chroma are not well-aligned with the functional descriptions. A possible explanation is that Chroma was trained on a significantly smaller number of caption-protein pairs compared to other models, resulting in inferior performance in understanding textual descriptions. In contrast, PRODVA consistently demonstrates the ability to design well-folded proteins with user-specified functions.

**(2) Incorporating additional data may potentially improve performance, particularly in terms of language alignment.** After fine-tuning ProteinDT and PAAG, these models demonstrate relatively

Table 2: Performance on the Mol-Instructions protein design task (**Best**, Second Best). "Pairs" refers to the text-protein pairs used during training and "Params" denotes the model's parameter size. Violin plots are provided in Appendix E.7. $^{\dagger}$ indicates that the training scripts are not publicly available.

| Models | #Pairs #Params | Sequence Plausibility | | Foldability | | | | Language Alignment (in %) | | | Sequence Diversity (↑) |
|---|---|---|---|---|---|---|---|---|---|---|---|
| | | PPL (↓) | Rep (↓) | pLDDT (↑) | % > 70 (↑) | PAE (↓) | % < 10 (↑) | ProTrek Score (↑) | EvoLlama Score (↑) | Retrieval Accuracy (↑) | |
| Natural | - | 318.15 | 0.02 | 80.64 | 81.27 | 9.20 | 65.73 | 27.00 | 60.33 | 84.85 | - |
| Random (U) | - | 2484.03 | 0.01 | 22.96 | 0.16 | 24.85 | 0.56 | 1.03 | 36.23 | 6.89 | 97.01 |
| Random (E) | - | 3136.88 | 0.01 | 25.77 | 0.20 | 24.71 | 0.60 | 1.04 | 34.11 | 6.78 | 99.56 |
| Random+ (E) | - | 846.01 | 0.01 | 64.47 | 37.03 | 17.91 | 7.52 | 0.30 | 38.65 | 6.13 | 98.63 |
| ProteinDT | 541K/729M | 1576.23 | 0.07 | 38.29 | 0.98 | 25.13 | 0.40 | 1.20 | 40.57 | 9.28 | **99.23** |
| ProteinDT$_{FT}$ | 712K/729M | 1213.38 | 0.04 | 51.42 | 25.61 | 18.57 | 23.92 | 13.89 | 52.84 | 47.29 | 79.87 |
| Pinal$^{\dagger}$ | 1.76B/2B | **308.97** | 0.13 | 75.25 | 68.97 | 10.96 | 58.44 | **17.50** | **53.42** | 57.95 | 82.96 |
| PAAG | 130K/1.3B | 2782.70 | 0.02 | 28.39 | 0.07 | 25.38 | 0.10 | 1.29 | 34.39 | 7.06 | 99.15 |
| PAAG$_{FT}$ | 712K/1.3B | 1332.35 | 0.04 | 50.37 | 23.86 | 19.96 | 21.99 | 10.04 | 49.69 | 33.66 | 86.09 |
| Chroma$^{\dagger}$ | 45K/334M | 1370.21 | 0.03 | 59.18 | 20.17 | 15.03 | 28.62 | 2.10 | 40.10 | 7.33 | 96.13 |
| PRODVA | 712K/1.8B | 415.63 | **0.02** | **76.86** | **76.35** | **8.66** | **68.06** | 17.40 | 51.10 | **59.07** | 83.29 |

improved performance across all metrics, with improvements of 12.69% and 8.75% on the ProTrek Score, and 12.27% and 15.30% on the EvoLlama Score, respectively. These results indicate that training with more data can significantly enhance a model's ability to understand textual functional descriptions written in natural language. In comparison, Pinal has been trained on approximately 1.76 billion text–protein pairs using both manually annotated and synthetic annotations, and demonstrates competitive performance on foldability and language alignment metrics. However, the large-scale datasets and training scripts used by Pinal have not yet been open-sourced. In contrast, our approach utilizes only 0.04% of the data and surpasses Pinal by generating 7.38% and 9.62% more structurally plausible proteins with pLDDT > 70 and PAE < 10, respectively. For language alignment, PRODVA also demonstrates highly comparable performance, being only marginally outperformed by Pinal with a gap of 0.1% on the ProTrek Score and 2.32% on the EvoLlama Score.

**(3) PRODVA demonstrates competitive sequence diversity compared to other baselines.** We find that baselines that struggle to design well-folded or well-aligned proteins consistently exhibit higher sequence diversity, comparable to that of random methods. Furthermore, textual descriptions containing more constraints than simple keywords generally result in lower sequence diversity, suggesting that complex function descriptions may restrict the design space. By leveraging the top-k sampling strategy during decoding, PRODVA achieves slightly better performance than Pinal on Mol-Instructions and surpasses it by 15.86% on the CAMEO subset.

### 4.4 Unconditional Protein Generation

Table 3: Performance on the unconditional protein generation task (**Best**, Second Best). The fine-tuned models and PRODVA are initialized with the weights obtained in Section 4.3, without additional fine-tuning for this task.

| Models | PPL (↓) | Rep (↓) | pLDDT (↑) | % > 70 (↑) | PAE (↓) | % < 10 (↑) |
|---|---|---|---|---|---|---|
| ProteinDT$_{FT}$ | 593.06 | 17.92 | 47.79 | 0.02 | 26.56 | 0.00 |
| PAAG$_{FT}$ | 1327.98 | 3.55 | 50.32 | 23.83 | 19.95 | 22.24 |
| Pinal | **411.93** | 14.05 | 70.11 | 57.02 | 12.76 | 48.44 |
| PRODVA | 476.02 | **1.47** | **77.52** | **79.78** | **9.32** | **60.25** |

Although PRODVA is designed for conditional protein design tasks, we demonstrate that it can be adapted for unconditional protein generation. By fixing the input instruction to `Design a novel protein sequence`, the problem addressed by PRODVA effectively transforms into an unconditional protein generation task. To further ensure that the input acts as a no-op, we replace the retrieval method with the random selection of fragments from the training set described in Section 4.3. For a fair comparison, each model generates the same number of protein sequences as the size of the Mol-Instructions test set. The results are shown in Table 3, PRODVA outperforms all baseline models on the unconditional protein generation task. Compared to the state-of-the-art baseline, Pinal, PRODVA generates 22.76% and 11.81% more well-folded proteins in terms of pLDDT and PAE, respectively. Furthermore, compared to other fine-tuned models, PRODVA achieves substantially superior performance. These findings highlight the effectiveness of our approach in both conditional and unconditional protein generation settings.

## 4.5 Comparison with a Vanilla Multimodal Approach

To further assess the effectiveness of our approach, we adopt methods [40, 41] commonly used in other fields, such as computer vision, to train a vanilla multimodal baseline. This baseline model integrates GPT-2 for text embedding, which serves as the input to ProtGPT2. Compared to PRODVA, all dynamic protein vocabulary components are excluded, making next token prediction the only learning objective. The baseline is trained using the same experimental setups described in Section 4.1 and the same dataset used in Section 4.3.

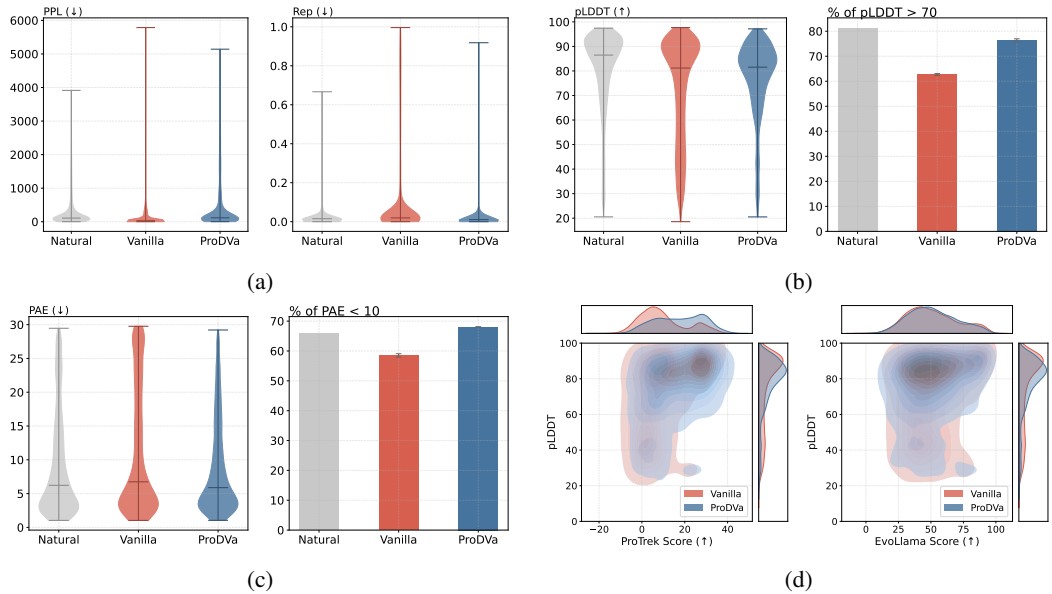

Figure 4: Experimental results comparing PRODVA with a vanilla multimodal baseline on Mol-Instructions. **(a)** illustrates performance on sequence plausibility metrics. **(b)** and **(c)** show performance on foldability metrics. **(d)** presents performance on language alignment metrics, where deeper colors in the upper right corner indicate better performance.

Figure 4 illustrates that the vanilla multimodal baseline outperforms most models presented in Table 2, indicating that even a simple integration of multimodal representations from textual descriptions can convey strong and detailed information. However, compared to PRODVA, the proteins designed by the baseline exhibit more repetitive patterns, resulting in a 4.63% decrease in pLDDT scores and a 2.71% increase in PAE scores. Furthermore, PRODVA achieves a 13.66% higher proportion of proteins with pLDDT > 70 and a 9.57% higher proportion with PAE < 10. These results underscore the importance of incorporating fragments and functional annotations, which enable the model to design proteins with more plausible structures. Furthermore, PRODVA significantly outperforms the baseline on the ProTrek Score with an average improvement of 7.77%, demonstrating its superior ability to design well-aligned proteins that more effectively meet specific user requirements.

## 4.6 Retrieving the Top $K$ Most Relevant Descriptions

The value of $K$ affects the size of the fragment candidate set during inference. Therefore, selecting an optimal $K$ has a significant impact on the model's performance. As shown in Figure 5, larger fragment candidate sets consistently lead to higher pLDDT scores, until more than 64 relevant descriptions are retrieved. Although the model's performance on PPL and PAE deteriorates with increasing $K$, the average PAE scores remain within a reasonable range (i.e., PAE < 10). For language alignment metrics, we observe that the ProTrek Score decreases when selecting more than the 16 most relevant descriptions. A possible explanation is that retrieving the top $K$ most relevant descriptions helps filter out irrelevant proteins and fragments, thereby enhancing the model's performance in designing user-specified proteins. These results demonstrate that simply increasing the number of fragment candidates during inference does not consistently yield positive outcomes, highlighting the

importance of selecting an optimal Top $K$ value. We provide a more detailed analysis of this pattern with additional experiments in Appendix E.9.

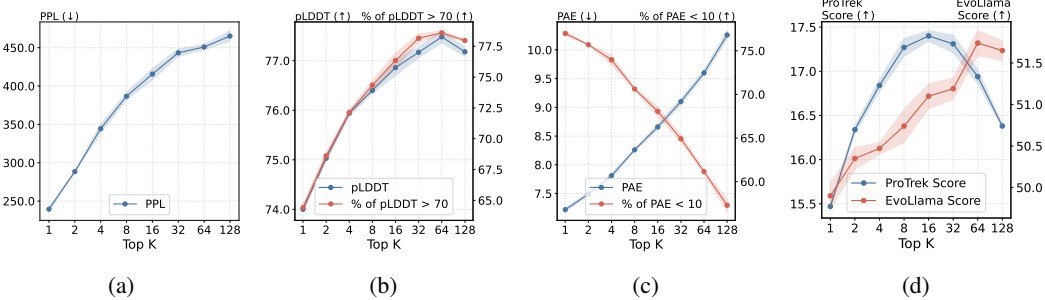

Figure 5: Analysis regarding the selection of Top $K$ most relevant descriptions during inference on Mol-Instructions. Results on the CAMEO subset are provided in Appendix E.8.

# 5 Related Work

The protein design problem can be broadly categorized into two paradigms: unconditional and conditional protein design. Unconditional methods aim to generate novel protein sequences or structures without specific constraints. Representative sequence-based approaches include DARK [42], RITA [43], ProtGPT2 [23], and DPLM [44], while structure-based methods include Multiflow [45] and DPLM-2 [46]. In contrast, conditional protein design seeks to generate proteins under desired functional or structural constraints [47, 48, 49, 50]. A key area of conditional generation is function-based protein design. Early models such as ProGen [4], ProGen2 [51], and ESM-3 [5] condition generation on control tags or multi-hot encodings. However, these approaches are limited in expressiveness and cannot fully interpret natural language. Recent methods, including ProteinDT [6], Pinal [8], PAAG [7], and Chroma [26], aim to support arbitrary textual functional descriptions. Despite this, they often underexploit rich functional annotations or employ less effective learning strategies, resulting in protein sequences that are poorly folded or misaligned with the user-specified functions. We discuss more related work in Appendix A.

# 6 Conclusion

In this paper, motivated by classical approaches to protein design, we propose a novel method, PRODVA, which integrates a text language model, a protein language model backbone, and a fragment encoder. On protein design tasks based on both function keywords and textual descriptions, our approach demonstrates competitive performance in language alignment and significantly outperforms state-of-the-art baselines in designing well-folded proteins.

## Acknowledgement

The authors wish to thank all reviewers for their helpful comments and suggestions. We also thank Jie Wang for his insightful discussions. The corresponding authors are Yuanbin Wu, Tao Ji, Changzhi Sun and Man Lan. The computations in this research were performed using the CFFF platform of Fudan University.

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

## A  More Related Work

Prior work on dynamic vocabulary [15, 16, 17] explores augmenting the fixed vocabulary of language models with phrases extracted from retrieved documents, typically using heuristics such as Forward Maximum Matching or N-gram extraction. While effective for natural language tasks, these approaches differ from ours in three key points. **First**, function-based protein design is a multimodal problem that requires aligning natural language descriptions with amino acid sequences, making cross-modal retrieval and generation more challenging. **Second**, prior methods extract phrases based on token-level or word-level heuristics, often resulting in semantically shallow units that serve mainly as grammatical components. **Third**, our method leverages InterPro [18] to identify biologically meaningful fragments with functional annotations, enabling more interpretable protein vocabulary construction, leading to better generation of user-specified protein sequences.

## B  Details of InterPro and Fragment Types

InterPro [18] is an integrated database and diagnostic tool that facilitates the functional analysis of proteins by classifying them into families and predicting domains and functional sites. It achieves this by utilizing predictive models from several member databases, such as Pfam [52]. While InterPro offers a user-friendly GUI for searching, in our experiments, we may utilize InterProScan [34], a software package that can be executed as a command-line tool to scan sequences against the InterPro database.

Table 4 presents the details of the 8 fragment types identified by InterPro.

Table 4: Descriptions of 8 fragment types identified by InterPro.

| Types | Descriptions |
|---|---|
| Domains | Distinct functional, structural or sequence units that may exist in a variety of biological contexts. |
| Family | A group of proteins that share a common evolutionary origin reflected by their related functions, similarities in sequence, or similar primary, secondary or tertiary structure. |
| Homologous Superfamily | A group of proteins that share a common evolutionary origin reflected by their related functions, similarities in sequence, or similar primary, secondary or tertiary structure. (usually predicted by Hidden Markov Models) |
| Repeat | A short sequence that is typically repeated within a protein. |
| Conserved Site | A short sequence that contains one or more conserved residues. |
| Active Site | A short sequence that contains one or more conserved residues, which allow the protein to bind to a ligand. |
| Binding Site | A short sequence that contains one or more conserved residues, which form a protein interatcion site. |
| PTM | A short sequence that contains one or more conserved residues. Post-Traditional Modification site. |

## C  Details of the Experimental Setups

### C.1  Training Details

PRODVA is trained using the AdamW optimizer [53] with $\beta_1 = 0.9$, $\beta_2 = 0.95$, and a gradient clipping of 1.0. The maximum learning rate is set to $1 \times 10^{-4}$, with a linear warmup over the first 5% of training steps. The learning rate remains constant after warmup and follows a "1-sqrt" decay schedule during the last 10% of training steps. The minibatch size is set to 4 to ensure memory efficiency, with an overall batch size of 64. During training, the weights of the Text Language Model are frozen. Training is conducted for 10K steps on the CAMEO subset and 20K steps on Mol-Instructions. The loss weights are set to $\alpha = \beta = 0.2$.

## C.2  Dataset Construction Details

Table 5: Statistics of the datasets.

| Datasets | #Training | #Validation | #Test |
|---|---|---|---|
| CAMEO subset [36] | 391,933 | 20,629 | 507 |
| SwissProtCLAP [6] | 541,159 | - | - |
| Mol-Instructions (Protein Design) [37] | 171,089 | 19,010 | 5,876 |

The statistics of the datasets used in Sections 4.2 and 4.3 are presented in Table 5.

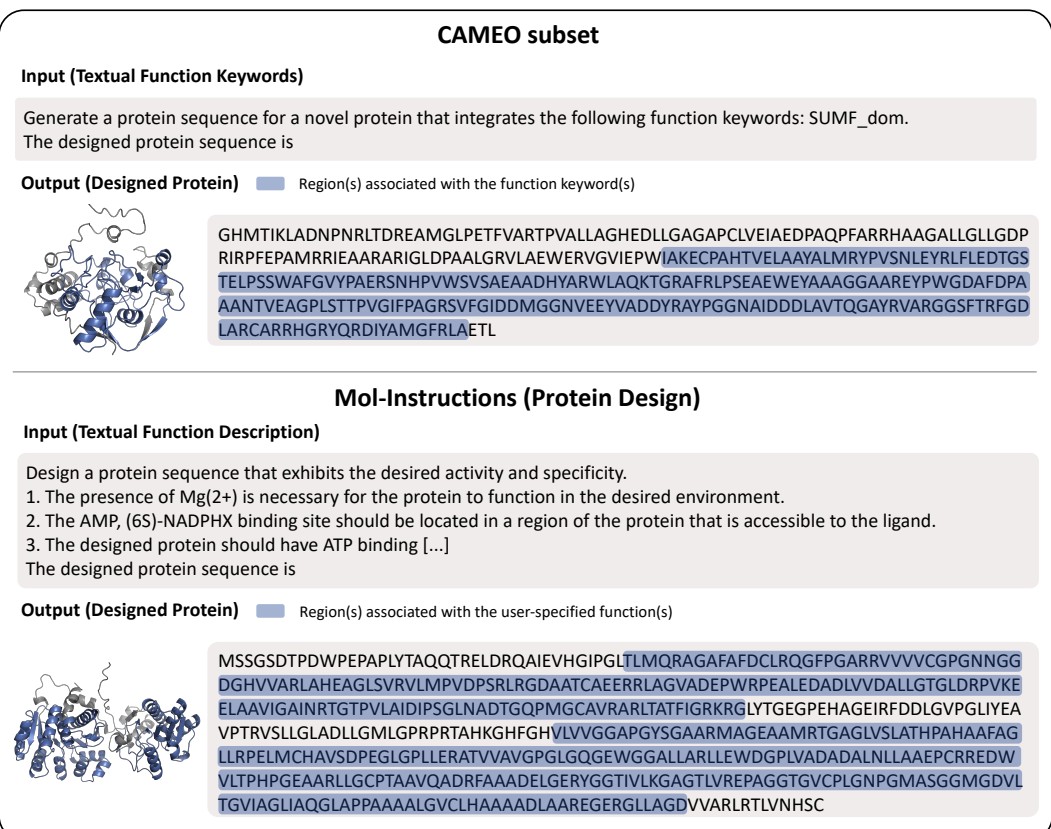

Figure 6: Examples from the CAMEO subset and Mol-Instructions. Regions associated with the input functions are highlighted in both the protein structures and sequences. These regions include specific domains or arbitrary domains related to user-specified functions.

The CAMEO subset consists of function keyword–protein pairs, where the function keywords are formulated as simple instructions tailored to our use cases. Each protein may be associated with one or more function keywords. In contrast, Mol-Instructions consists of instruction–protein pairs, where each instruction may include one or more diverse functional descriptions written in natural language. Examples of these two datasets are illustrated in Figure 6.

# D  Details of the Metrics

In this section, we present detailed definitions and implementations of the metrics introduced in Section 4.1.

## D.1 Sequence Plausibility Metrics

**Perplexity** Perplexity (PPL) is a commonly used metric in Natural Language Processing to evaluate the quality of sentences generated by a language model. In our experiments, we use PPL to assess how effectively a model has learned intricate patterns in protein sequences and to provide a rapid validation of protein structures. Lower PPL generally indicates a more accurately folded structure. Specifically, we employ ProtGPT2 [23] to evaluate the PPL as follows:

$$\text{PPL} = \exp\left(-\frac{1}{n}\sum_{i=1}^{n}\log p(x_i \mid x_{<i})\right)$$

where $(x_1, x_2, \cdots, x_n)$ is a sequence of tokens tokenized using the ProtGPT2 tokenizer. Additional PPL evaluations using alternative language models are provided in Appendix E.2.

**Repetitiveness** Some previous studies [23, 44] have suggested that repetitive patterns in amino acid sequences may result in regions of intrinsic disorder, which are typically associated with lower pLDDT scores. Inspired by Rep-N [27], which measures repetition at the N-gram level, we adopt Repetitiveness, a metric that calculates the proportion of repeated subsequences within the entire sequence.

## D.2 Foldability Metrics

**pLDDT** The predicted Local Distance Difference Test (pLDDT) is a per-residue evaluation metric that produces a pLDDT score array of size $\mathbb{R}^{1 \times n}$, where $n$ denotes the length of the amino acid sequence. Each score in the array ranges from 0 to 100 and estimates the agreement between the predicted and experimental structures. It is based on the Local Distance Difference Test $C\alpha$ [54], which evaluates the correctness of the local atomic distances without requiring superposition. We report the average pLDDT score across all residues.

**PAE** The Predicted Aligned Error (PAE), computed by ESMFold, quantifies the model's confidence in the relative positioning of residue pairs within a predicted structure. It represents the expected positional error, in Å, at residue X when the predicted and actual structures are aligned on residue Y, thereby indicating the reliability of domain packing and inter-domain arrangement. Consequently, the PAE is represented as a matrix of size $\mathbb{R}^{n \times n}$, where $n$ is the length of the amino acid sequence. We report the average PAE score across this matrix.

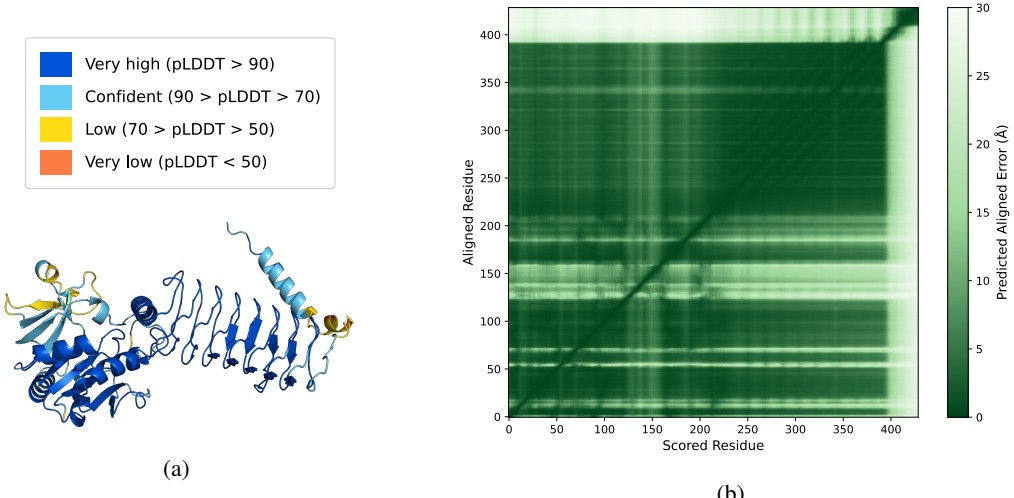

(a)

(b)

Figure 7: An example of G9WEQ2. **(a)** The pLDDT score of the protein, where blue regions indicate well-folded areas. **(b)** The PAE matrix of the protein, with darker green indicating higher predicted structural correctness (i.e., lower predicted error).

### D.3 Language Alignment Metrics

**ProTrek Score**  The ProTrek Score is defined as the cosine similarity between a textual function description $t$ and a designed protein sequence $P$. Since PRODVA generates protein sequences directly, without an intermediate step for designing structure backbones, only the text and sequence encoders of ProTrek [32] are utilized. Specifically, we employ the pre-trained ProTrek 650M [3] as the oracle model. The ProTrek Score is formally defined as:

$$\text{ProTrek Score} = \text{sim}(\tau_t(t), \tau_p(P))$$

where $\text{sim}(\cdot, \cdot)$ denotes the cosine similarity function, and $\tau_t(t)$ and $\tau_p(P)$ represent the embeddings of the textual description and the protein sequence, respectively. Note that we follow the original implementation of the ProTrek Score [4] as described in [8]. In contrast, the latest version of the paper defines the ProTrek Score with an additional division by the model temperature.

**EvoLlama Score**  Unlike the ProTrek Score, which is directly defined based on the similarity between the function description $t$ and the protein sequence $P$, we propose a generative approach to compute the alignment using EvoLlama [33]. Specifically, we adopt EvoLlama with a 650M ESM2 protein sequence encoder and a 3B Llama-3.2 text decoder. The model is randomly initialized and trained from scratch using the Mol-Instructions *de novo* design datasets described in Section 4.3. During evaluation, the fine-tuned EvoLlama takes the protein sequence $P$ and a simple prompt, `The function of the protein is`, as inputs and generates a predicted function description $t'$. Assume that the ground truth description $t$ and the generated description $t'$ can be tokenized into $k$ and $k'$ tokens, respectively. The EvoLlama Score is then defined as:

$$\text{EvoLlama Score} = \text{sim}\left(\frac{1}{k}\sum_{i=1}^{k}\texttt{Embed}(t), \frac{1}{k'}\sum_{i=1}^{k'}\texttt{Embed}(t')\right)$$

where $\texttt{Embed}(\cdot)$ denotes using PubMedBERT [24] as the embedding model.

**Keyword Recovery**  ESM3 [5] lacks the capability to understand arbitrary natural languages, limiting it to accepting only multi-hot function keywords as inputs. Keyword Recovery is a protein-level metric that measures the percentage of function keywords identified in the designed proteins using InterProScan [34]. Let the reference set of function keywords (ground truth) be denoted as $K_{\text{ref}}$, and the set of function keywords associated with the designed sequence be denoted as $K_{\text{pred}}$. The Keyword Recovery metric can then be defined as follows:

$$\text{Keyword Recovery} = \frac{|K_{\text{pred}} \cap K_{\text{ref}}|}{|K_{\text{ref}}|}$$

**Retrieval Accuracy**  Retrieval Accuracy [6] evaluates a model's ability to correctly associate a textual function description $t$ with its corresponding protein sequence $P$. It is computed by comparing the similarity between the text description embedding and multiple protein embeddings, including one correct match and $T - 1$ randomly retrieved negative samples. The metric is defined as the proportion of instances in which the correct protein sequence exhibits the highest similarity to the function description among all candidates. Note that the original implementation relies on ProteinCLAP [6] for both text and protein embeddings, whereas we adopt ProTrek [32] to achieve faster and more accurate representations, denoted as $\texttt{Embed}$. In our evaluation, $T$ is set to 20, representing the most challenging setting in [6]. The Retrieval Accuracy for a text-protein pair $(t, P_1)$ is defined as:

$$\text{Retrieval Accuracy@}T = \mathbf{1}\left[\text{sim}\left(\texttt{Embed}(t), \texttt{Embed}(P_1)\right) = \max_{j=1}^{T}\left\{\text{sim}\left(\texttt{Embed}(t), \texttt{Embed}(P_j)\right)\right\}\right]$$

where $\mathbf{1}(\cdot)$ is the indicator function.

---

[3] `https://huggingface.co/westlake-repl/ProTrek_650M_UniRef50`
[4] August 2024 version: `https://www.biorxiv.org/content/10.1101/2024.08.01.606258v1.full.pdf`

### D.4 Sequence Diversity Metric

**Sequence Diversity**  Sequence Diversity measures a model's ability to generate diverse protein sequences under identical conditions (i.e., given the same function as inputs). Given a textual description, the model is tasked with designing a batch of $N$ protein sequences, denoted as $\{P_1, P_2, \cdots, P_N\}$. Formally, Sequence Diversity is defined as:

$$\text{Sequence Diversity} = \frac{2}{N(N-1)} \sum_{1 \leq i \leq j \leq N} \text{sim}(P_i, P_j)$$

where MMseqs2 [35] is used to compute the sequence-level similarity, $\text{sim}(\cdot, \cdot)$, between each pair of proteins $P_i$ and $P_j$.

## E  Additional Evaluations

### E.1  Additional UMAP Visualization of Designed Proteins

In Figure 1, we present a visualization of proteins embedded using ESM C, a state-of-the-art foundation model designed for computing protein representations. To further validate the intuition behind PRODVA, we utilize Pinal to compute the protein representations. Although Pinal, like PRODVA, is not specifically designed for protein representation, it consists of two primary modules: a text-to-structure module and a SaProt [55] module, the latter of which can serve as an embedding model. The weights of the SaProt module are initialized using the Pinal checkpoint [5].

Following the approach used in Figure 1(a), the additional UMAP visualization is presented in Figure 8. The results are highly consistent with those obtained using ESM C. Proteins generated by `Random` form distinct clusters, while those from `Random+` exhibit a significantly more diverse distribution. As expected, proteins designed by PRODVA span the landscape of natural proteins.

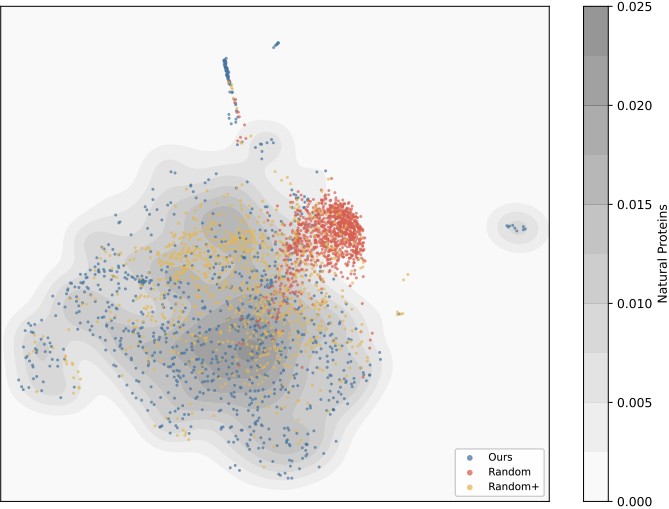

Figure 8: Additional visualization of proteins designed by our method, `Random`, and `Random+`, embedded using Pinal (SaProt module) and projected with UMAP.

### E.2  PPL Evaluation Using Alternative Protein Language Models

To provide a more comprehensive evaluation of the PPL metric, we select two additional language models (i.e., ProGen2 [51] and RITA [43]) pre-trained on a large scale of protein sequences. The evaluations conducted on Mol-Instructions are displayed in Table 6. The results are consistent with the PPL values computed by ProtGPT2, despite differences in absolute scores. PRODVA consistently remains within a low PPL range and is closer to natural proteins.

---

[5] `https://huggingface.co/westlake-repl/Pinal/tree/main/SaProt-T`

Table 6: Comparison of PPL performance computed using ProtGPT2, ProGen2, and RITA.

| Models | PPL$_{\text{ProtGPT2}}$ ($\downarrow$) | PPL$_{\text{ProGen2}}$ ($\downarrow$) | PPL$_{\text{RITA}}$ ($\downarrow$) |
|---|---|---|---|
| Natural | 318.15 | 5.99 | 5.52 |
| Random (U) | 2484.03 | 21.71 | 22.14 |
| Random (E) | 3136.88 | 18.68 | 19.04 |
| Random+ (E) | 846.01 | 10.08 | 9.32 |
| ProteinDT | 1576.23 | 12.41 | 12.44 |
| ProteinDT$_{\text{FT}}$ | 1213.38 | 10.80 | 10.69 |
| Pinal | **308.97** | **5.81** | **5.77** |
| PAAG | 2782.70 | 17.84 | 18.05 |
| PAAG$_{\text{FT}}$ | 1332.35 | 11.09 | 11.03 |
| Chroma | 1370.21 | 12.22 | 12.42 |
| PRODVA | 415.63 | 7.63 | 8.82 |

## E.3 Further Analysis of Foldability Metrics

The pLDDT and PAE metrics in our experiments are averaged across all designed proteins. To mitigate the potential bias of foldability metrics being dominated by a few clusters, we first exclude well-folded proteins. The remaining proteins are then clustered using MMseqs2 based on sequence identity. Finally, the average pLDDT and PAE are computed across the filtered clusters. Assuming the filtered proteins are grouped into $k$ clusters, the metric can be defined as follows:

$$\text{pLDDT}_{\text{clusters}} = \frac{1}{k} \sum_{j=1}^{k} \left( \frac{1}{n_j} \sum_{i=1}^{n_j} \text{pLDDT}_{ij} \right)$$

where $n_j$ denotes the number of proteins in the $j$-th cluster. The PAE$_{\text{clusters}}$ can be defined in a similar way. The results are shown in Table 7. For these filtered proteins that are not well-folded (pLDDT below 70 or PAE above 10), PRODVA outperforms other models by a substantial margin, with average scores across clusters approaching the well-folded threshold. This highlights the robustness of our method in designing structurally plausible proteins.

Table 7: The average pLDDT and PAE scores across clusters (denoted as pLDDT$_{\text{clusters}}$ and PAE$_{\text{clusters}}$, respectively) are evaluated on Mol-Instructions. The percentage represents the sequence identity threshold. For clarity of comparison, the results for unfiltered proteins (i.e., including well-folded proteins) are presented in the parentheses.

| Models | pLDDT ($\uparrow$) | pLDDT$_{\text{clusters}}$ ($\uparrow$) | | PAE ($\downarrow$) | PAE$_{\text{clusters}}$ ($\downarrow$) | |
|---|---|---|---|---|---|---|
| | | 30% | 50% | | 30% | 50% |
| Natural | 80.64 | 67.04 (77.24) | 67.79 (80.44) | 9.20 | 17.83 (11.25) | 17.59 (9.61) |
| Random (U) | 22.96 | 22.87 (22.96) | 22.87 (22.96) | 24.85 | 24.88 (24.85) | 24.88 (24.85) |
| Random (E) | 25.77 | 25.68 (25.77) | 25.68 (25.77) | 24.71 | 24.74 (24.71) | 24.74 (24.71) |
| Random+ (E) | 64.47 | 63.19 (64.32) | 63.20 (64.40) | 17.91 | 18.73 (18.01) | 18.72 (17.96) |
| ProteinDT | 38.29 | 38.04 (38.10) | 38.10 (38.16) | 25.13 | 25.22 (25.19) | 25.21 (25.19) |
| ProteinDT$_{\text{FT}}$ | 51.42 | 39.47 (40.53) | 40.17 (43.57) | 18.57 | 22.93 (22.50) | 22.73 (21.41) |
| Pinal | 75.25 | 57.71 (66.90) | 57.59 (67.87) | 10.96 | 19.57 (14.81) | 19.73 (14.41) |
| PAAG | 28.39 | 28.37 (28.39) | 28.37 (28.39) | 25.38 | 25.39 (25.38) | 25.39 (25.38) |
| PAAG$_{\text{FT}}$ | 50.37 | 39.14 (39.93) | 39.87 (43.17) | 19.96 | 24.48 (24.14) | 24.23 (22.85) |
| Chroma | 59.18 | 55.36 (59.13) | 55.37 (59.17) | 15.03 | 17.13 (15.06) | 17.13 (15.03) |
| PRODVA | **76.86** | **64.59** (74.51) | **66.49** (76.68) | **8.66** | **14.19** (9.09) | **14.37** (8.74) |

### E.4 Effect of Loss Weighting

As discussed in Section 3.2, $\alpha$ and $\beta$ are hyperparameters adjusting the loss weights. To determine their optimal values, we conduct experiments on the validation set of Mol-Instructions. Figure 9 demonstrates that incorporating both $\mathcal{L}_{\text{TYPE}}$ and $\mathcal{L}_{\text{DESC}}$ significantly improves the model's performance across all metrics. Specifically, we observe that continuously increasing the weight of either $\alpha$ or $\beta$ leads to decreased pLDDT and ProTrek Score. To balance foldability and language alignment, we further explore loss weights around 0.1 and 0.2, ultimately selecting $\alpha = \beta = 0.2$ to achieve an optimal trade-off.

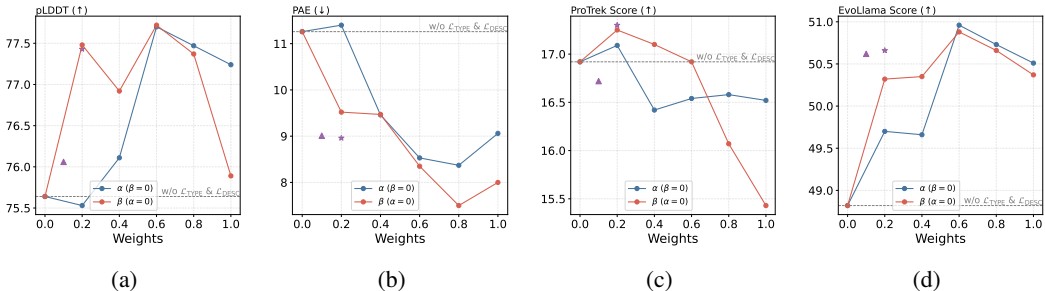

(a)  (b)  (c)  (d)

Figure 9: Evaluations on the effect of loss weighting. The blue lines indicate the effect of $\alpha$ with $\beta = 0$, while the red lines indicate the effect of $\beta$ with $\alpha = 0$. The triangle and star markers indicate configurations where both $\alpha$ and $\beta$ are set to 0.1 and 0.2, respectively. The grey dashed line denotes the performance without incorporating type loss and description loss during training.

### E.5 Varying the Size of the Supporting Documents

The size of the supporting documents constrains the number of text–protein pairs that can be retrieved or utilized during inference. To investigate the effect of varying the supporting document size, we present experimental results obtained by both decreasing and increasing the size of the supporting documents, as shown in Figure 10. The size of the supporting documents appears to have minimal impact on PPL and foldability metrics, with no significant differences observed as the size increases. In contrast, for the language alignment metrics, PRODVA generally performs better with larger supporting document sizes. A possible explanation is that a larger set of supporting documents increases the likelihood of retrieving more relevant descriptions and sequences, thereby facilitating the construction of more relevant fragment candidates.

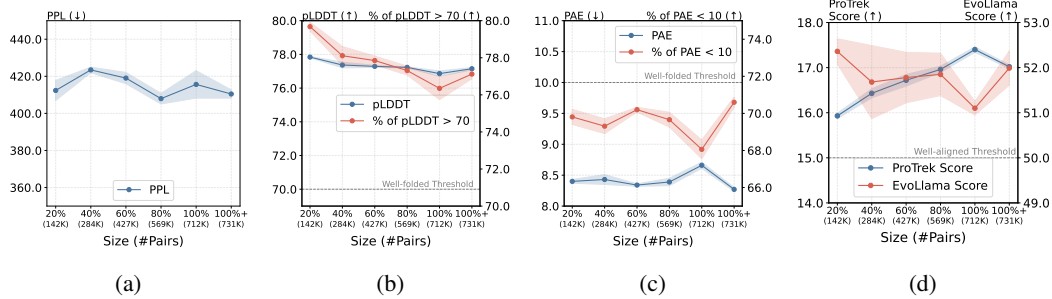

(a)  (b)  (c)  (d)

Figure 10: Analysis of the variation in supporting document size during inference on Mol-Instructions. "Size" denotes the proportion of the training set used as supporting documents, with 100% (i.e., the complete training set) serving as the default in Table 2. Additionally, the validation set is included to further expand the supporting document size, indicated as 100%+.

### E.6 Additional Evaluations on the CAMEO Subset

Violin plots illustrating the results on the CAMEO subset are shown in Figure 11. PRODVA and Pinal consistently outperform the baseline models across most metrics. Specifically, the majority of

proteins designed by our approach remain within a reasonable PPL range. Furthermore, PRODVA generates a greater number of well-folded proteins compared to Pinal, demonstrating the effectiveness of our method in designing structurally plausible proteins. Additionally, PRODVA demonstrates a distribution closely matching that of Pinal on both the ProTrek Score and Keyword Recovery metrics, despite using only 0.02% of the training data.

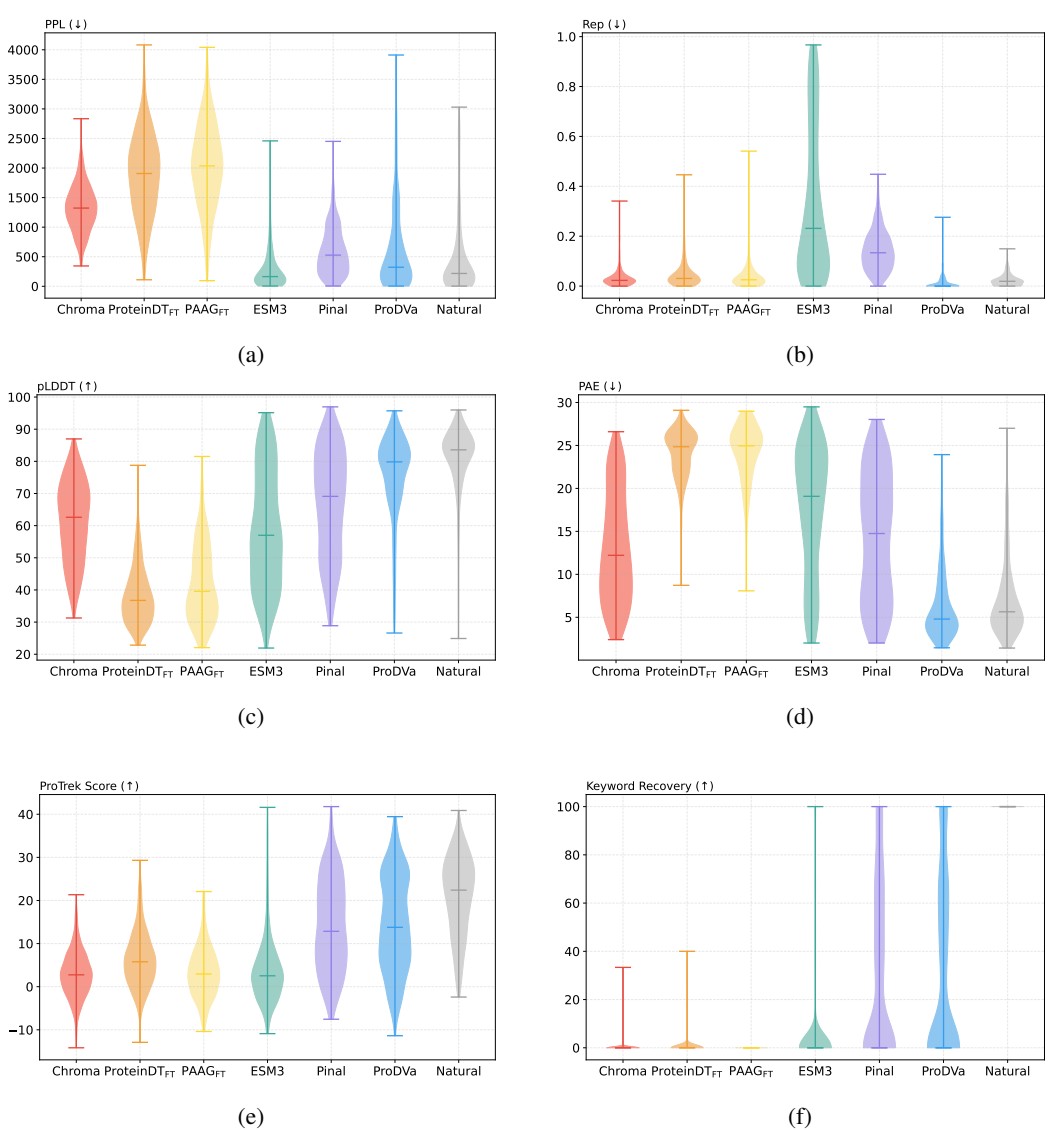

Figure 11: Violin plots illustrating the results on the CAMEO subset.

## E.7 Additional Evaluations on Mol-Instructions

Violin plots illustrating the results on the Mol-Instructions protein design task are shown in Figure 12. A similar conclusion can be drawn that PRODVA demonstrates consistent capability in designing well-folded and well-aligned proteins compared to other baseline models, including the state-of-the-art Pinal.

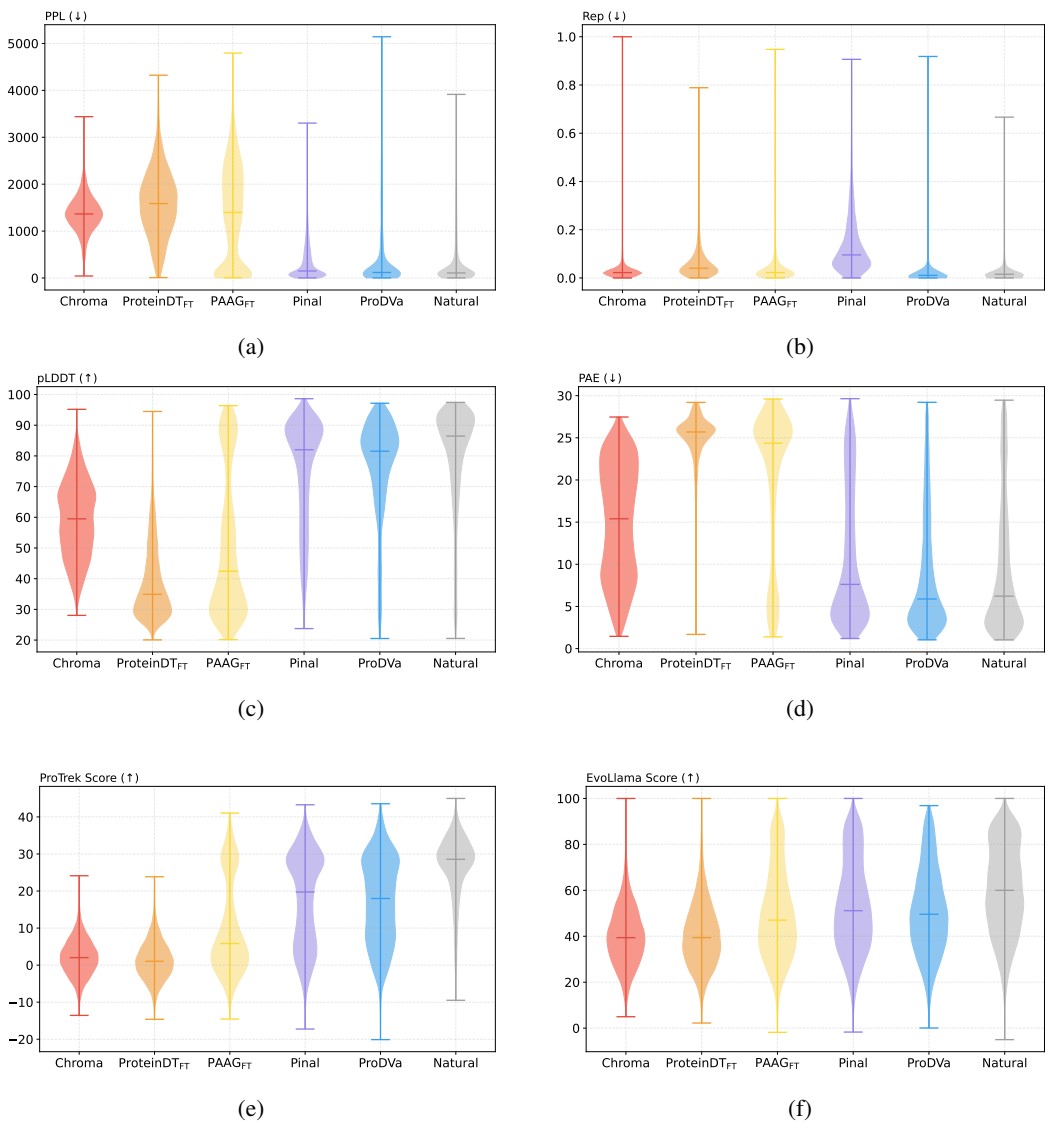

Figure 12: Violin plots illustrating the results on the Mol-Instructions protein design task.

## E.8 Additional Results on Retrieving the Top $K$ Most Relevant Descriptions

Further analysis is conducted on the CAMEO subset to determine the optimal value of $K$ for retrieval during inference. The experimental results, illustrated in Figure 13, exhibit a consistent pattern with the findings discussed in Section 4.6. Both PPL and PAE scores increase with larger values of $K$, yet remain within an acceptable range. The model's performance measured by the pLDDT score, initially improves but declines when more than 16 of the most relevant descriptions are retrieved. Additionally, the ProTrek Score achieves its highest performance when retrieving the top 16 most relevant descriptions. To balance well-folded structures and well-aligned proteins, we select $K = 16$ and $K = 32$ throughout all experiments.

## E.9 Inference with Additional Fragments

In Section 4.6, we argue that retrieving the top $K$ most relevant descriptions helps filter out irrelevant proteins and fragments, thereby improving the model's performance. To further investigate this, we conduct two additional experiments. The first involves incorporating randomly sampled fragments

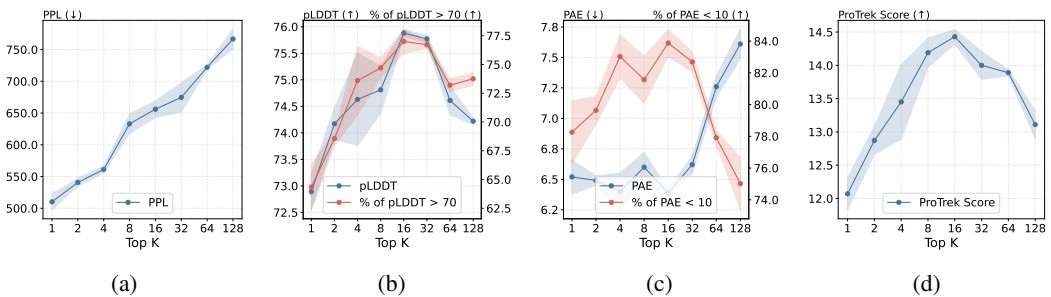

(a)  (b)  (c)  (d)

Figure 13: Analysis regarding the selection of Top $K$ most relevant descriptions during inference on the CAMEO subset.

that are not among those from the top $K$ relevant protein sequences. The second is to add N-gram subsequences of random lengths. Both cases can be considered as introducing noise either through additional irrelevant fragments or randomly generated N-gram subsequences.

Figure 14 presents several interesting findings, summarized as follows:

**(1) The Fragment Encoder learns to differentiate between biologically meaningful fragments and biologically meaningless N-gram subsequences.** The results in Figures 14(g) and (h) demonstrate that incorporating additional N-gram subsequences has marginal negative impact on both the ProTrek Score and the EvoLlama Score. In contrast, incorporating irrelevant fragments during inference significantly decreases the model's performance on language alignment. This indicates that our Fragment Encoder exhibits robustness to N-gram noise.

**(2) The retrieval method further filters out portions of the fragment noise.** Figures 14(a)–(f) show results consistent with those in Figures 5 and 13, indicating that introducing additional noise during inference has minimal impact on designing structurally plausible proteins. Therefore, the retrieval method effectively filters out irrelevant fragments as noise.

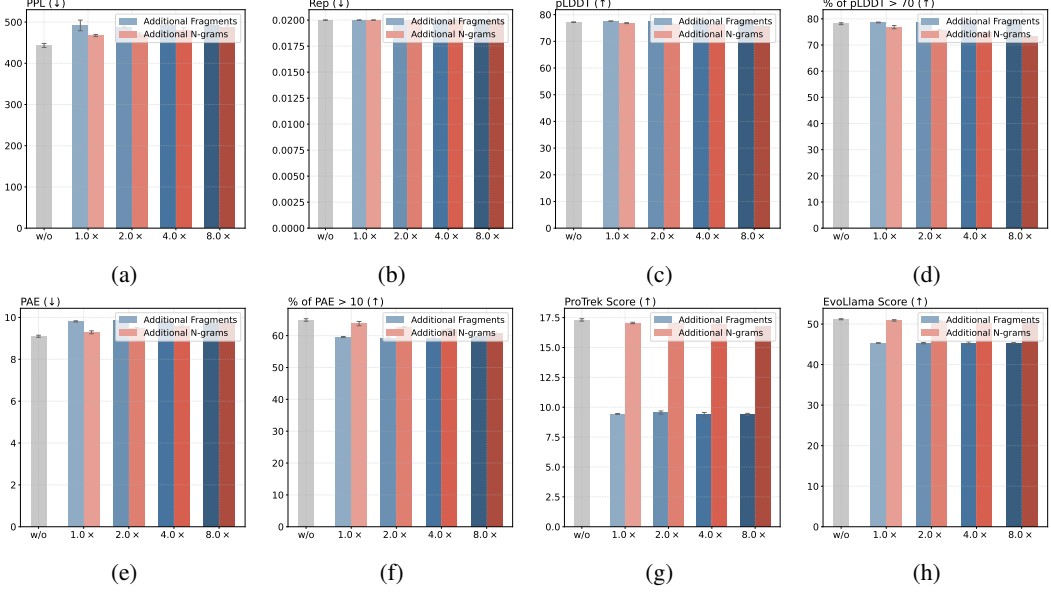

(a)  (b)  (c)  (d)

(e)  (f)  (g)  (h)

Figure 14: Results of incorporating additional fragments or N-gram subsequences during inference. The experiments are conducted under the setting of $K = 32$. Gray bars indicate the performance of retrieving the top 32 relevant descriptions without incorporating any additional fragments or N-grams. The x-axis represents the multiple of noise introduced relative to the original number of fragments during inference.

To further validate our first finding regarding the Fragment Encoder, we utilize UMAP [13] in Figure 15 to visualize the embeddings of fragments and N-grams represented by the Fragment Encoder. Additionally, we train a linear probe to classify the fragment and N-gram embeddings. The results demonstrate that, in a high-dimensional embedding space, a simple linear probe can accurately distinguish between the two types of subsequences, achieving an average classification accuracy of 93.87%. These findings support our conclusion that the Fragment Encoder effectively learns to differentiate between N-grams and fragments.

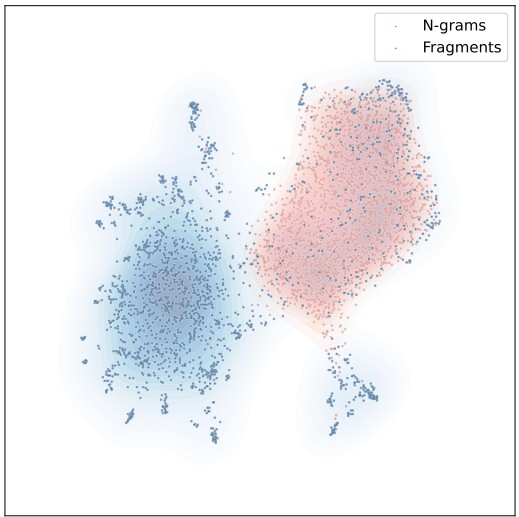

Figure 15: Visualization of fragment and N-gram embeddings represented by the Fragment Encoder.

## F  Case Study

An example of a protein designed by PRODVA is shown in Figure 16. PRODVA successfully designs a MobA-like NTP transferase domain that meets the input specifications. Additionally, the GlmU domain is designed to exhibit functions associated with the UDP-N-acetyl-alpha-D-glucosamine biosynthesis pathway.

## G  Limitations

In this paper, we employ four widely adopted language alignment metrics, including both oracle model-based and retrieval-based approaches, to evaluate the alignment between the designed proteins and their functional descriptions. While these metrics offer valuable insights and have been commonly used in prior work, they may not fully capture the biological validity or functional efficacy of the designed proteins. To address this limitation, future work will focus on conducting wet-lab experiments to empirically validate the functions of the designed proteins.

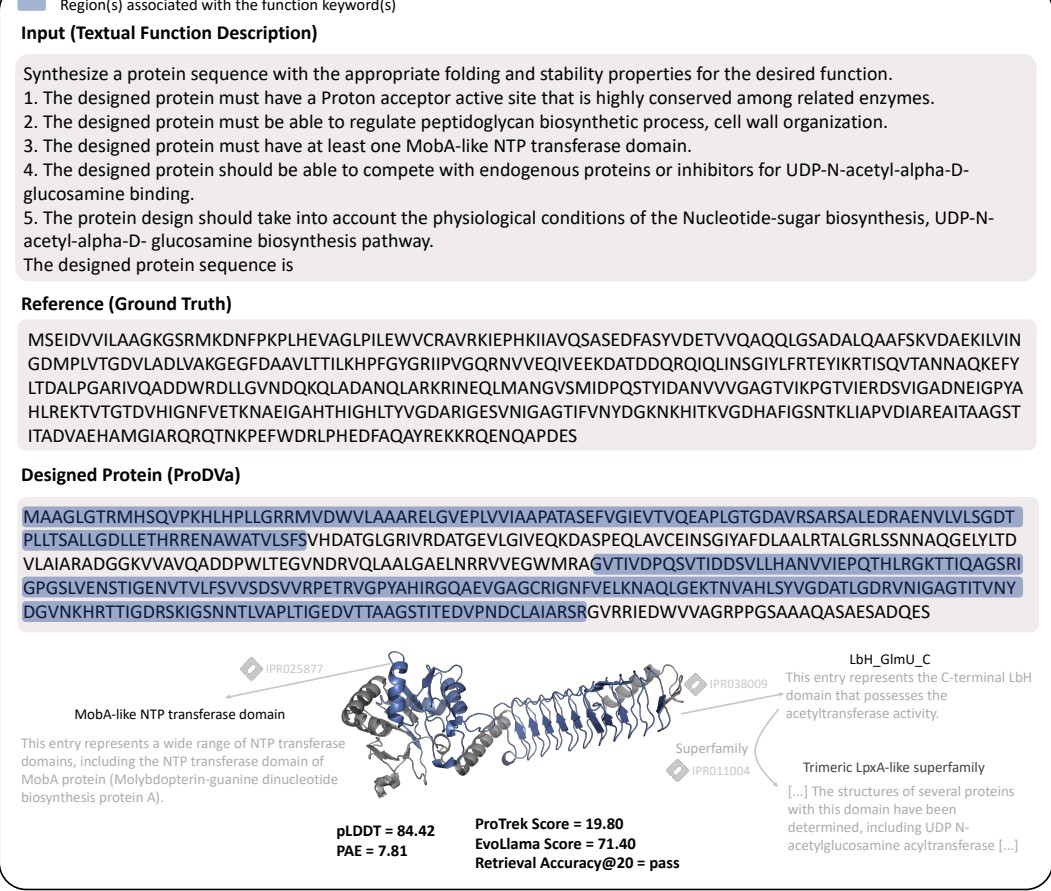

Figure 16: An example of a protein designed by PRODVA. The annotations are identified using InterPro, with two selected for illustration.

