# OpenReview forum: "Protein Design with Dynamic Protein Vocabulary"
_NeurIPS.cc/2025/Conference — NeurIPS 2025 spotlight_

### Official Review · Reviewer_Vivg · 2025-06-27

**Clarity:** 3
**Significance:** 3
**Originality:** 3
**Rating:** 5
**Confidence:** 4

**Summary:**

This paper proposes PRODVA, a protein sequence generation model based on a dynamic vocabulary of protein fragments. Specifically, the model constructs a database of fragment candidates. During training, it aligns the representations of a text language model and a fragment encoder on paired text-fragment data. During generation, the model is required to utilize the dynamically constructed fragments (in a manner similar to CopyNet[1] in text generation). In the inference stage, relevant fragments are retrieved based on the input text description and incorporated into the generation process.


[1] Gu, Jiatao, et al. "Incorporating copying mechanism in sequence-to-sequence learning." arXiv preprint arXiv:1603.06393 (2016).

**Questions:**

1. Please provide more ablation studies to validate the effectiveness of the model design. Specifically:

   * What is the effect of removing the dynamic vocabulary mechanism while still allowing the model to access top-k fragment candidates selected using the text encoder from a static fragment pool?
   * What is the effect of removing the fragment retrieval mechanism entirely (i.e., without any top-k fragment candidates)?
   * How does varying the size of the Fragment Candidates database affect model performance?

2. Recent diffusion-based language models (e.g., DPLM[1]) have shown strong potential in protein design. Can PRODVA’s approach be applied or adapted to diffusion models?

3. Since PRODVA relies on aligning textual descriptions with known fragment candidates, how well does it generalize to low-resource or novel protein design scenarios where such fragment annotations are unavailable or sparse?


[1] Wang, Xinyou, et al. "Diffusion language models are versatile protein learners." arXiv preprint arXiv:2402.18567 (2024).

**Ethical Concerns:**

["NO or VERY MINOR ethics concerns only"]

**Final Justification:**

In their latest response, they provided additional ablation study results, which demonstrate the rationality and effectiveness of the model architecture. This was my primary concern previously, and it has now been addressed. Therefore, I believe the paper can be accepted.

**Limitations:**

yes

**Quality:**

3

**Strengths And Weaknesses:**

Strengths

1. The paper addresses an interesting and meaningful problem—how to efficiently leverage protein fragment libraries for protein design, which is both non-trivial and highly relevant.
2. The writing is clear and well-structured, making the paper easy to follow.
3. PRODVA demonstrates strong performance across multiple tasks, showcasing the potential of this class of models.

Weaknesses

1. The paper lacks key ablation studies to justify and validate its design choices.
2. The method relies on text-based conditioning, which may limit its applicability in settings like unconditional protein generation.

---

> ### Author Rebuttal · Authors · 2025-07-31
>
> Dear Reviewer Vivg,
>
> First and foremost, we extend our gratitude for your thorough review. Here are our responses to the key points you've raised:
>
> Q1: More ablation studies.
>
> A1: Thanks for your valuable feedback.
>
> > What is the effect of removing the dynamic vocabulary mechanism while still allowing the model to access top-k fragment candidates selected using the text encoder from a static fragment pool?
>
> As shown in Figure 3, ProDVa relies on both the static (tokens) and dynamic (fragments) protein vocabulary to determine: 1) whether to output a token or fragment next, and 2) which token or fragment to select. Therefore, without the dynamic vocabulary mechanism, these two mechanisms cannot operate as intended. In other words, the static fragment pool cannot be utilized by ProDVa without the dynamic vocabulary mechanism.
>
> > What is the effect of removing the fragment retrieval mechanism entirely (i.e., without any top-k fragment candidates)?
>
> The table below compares two scenarios (i.e., the vanilla multimodal approach and the Top $K=0$ setting) that removing the fragment retrieval mechanism entirely. For further details on the vanilla multimodal approach, please refer to Section 4.4 in our manuscript.
>
> | Models  | Top K |  PPL $\downarrow$   | RepN  $\downarrow$ | PLDDT $\uparrow$ |  PAE  $\downarrow$  | ProTrek Score $\uparrow$ | EvoLlama Score  $\uparrow$ |
> | ------- | :---: | :----: | :--: | :---: | :---: | :-----------: | :------------: |
> | Vanilla |   -   | 109.66 | 0.10 | 72.23 | 11.37 |     9.63      |     50.99      |
> | ProDVa  |   0   | 107.63 | 6.25 | 62.83 | 7.89  |     3.80      |     40.77      |
> | ProDVa  |   1   | 239.52 | 0.03 | 74.00 | 9.22  |     15.47     |     49.90      |
> | ProDVa  |  16   | 415.63 | 0.02 | 76.86 | 8.66  |     17.40     |     51.10      |
>
> The results indicate that, compared to the vanilla multimodal approach, ProDVa without the use of any Top $K$ fragments performs worse across most metrics. However, when utilizing the Top 1 fragments, ProDVa improves its performance by 11.17% in pLDDT and 11.67% in the ProTrek Score. This highlights the significance of incorporating Top $K$ fragments. **The relatively poor performance without any Top $K$ fragments can be attributed to the fact that, during training, most sequences are split into tokens and fragments. Thus, relying solely on tokens (without incorporating any fragments) may lead to failure.**
>
> > How does varying the size of the Fragment Candidates database affect model performance?
>
> In our manuscript, the size of the supporting documents is identical to that of the training dataset for a fair comparison (see Lines 140–141). To investigate the effect of varying the size of the supporting documents (fragment database), we retain 20%, 40%, 60%, and 80% of the training dataset as supporting documents. Additionally, we incorporate the validation set to further increase the size of the supporting documents. For the statistics of the training and validation sets, please refer to Appendix C.2.
>
> The table below presents the experimental results obtained by decreasing or increasing the size of the supporting documents. "T" denotes the training set (by default, "100% T" is used in our manuscript, see Table 2), and "V" denotes the validation set. We summarize the key observations as follows:
>
> - **The size of the supporting documents has minimal impact on PPL and foldability, with no significant differences observed as the size increases.**
> - **Regarding the language alignment metrics, ProDVa generally performs better as the size of the supporting documents increases.** A possible explanation is that a larger set of supporting documents enhances the likelihood of retrieving more relevant descriptions and corresponding sequences, thereby enabling the construction of more relevant fragment candidates.
>
> |   Size (#pairs)   |  PPL $\downarrow$  | pLDDT $\uparrow$ | PAE $\downarrow$  | ProTrek Score $\uparrow$ | EvoLlama Score $\uparrow$ |
> | :---------------: | :----: | :---: | :--: | :-----------: | :------------: |
> |   20% T (142K)    | 412.37 | 77.84 | 8.40 |     15.93     |     52.36      |
> |   40% T (284K)    | 423.45 | 77.37 | 8.43 |     16.43     |     51.68      |
> |   60% T (427K)    | 419.04 | 77.29 | 8.34 |     16.72     |     51.78      |
> |   80% T (569K)    | 407.95 | 77.23 | 8.39 |     16.96     |     51.85      |
> |   100% T (712K)   | 415.63 | 76.86 | 8.66 |     17.40     |     51.10      |
> | 100% T + V (731K) | 410.46 | 77.15 | 8.27 |     17.02     |     51.99      |
>
> ---
>
> Q2: Clarifications on function-based protein design problems (i.e., conditional generation).
>
> A2: Sorry for misunderstanding. In our manuscript, we state that the problem addressed by ProDVa is function-based protein design (i.e., conditional generation). This problem can be formalized as follows (see Section 2, Lines 66–69):
>
> $$
> p(P \mid t) = p((x_1, x_2, \cdots, x_k) \mid t, \forall i, x_i \in A \cup S)
> $$
>
> where $t$ denotes the input textual description (i.e., the condition).
>
> Here, we provide a more detailed clarification as to why ProDVa is specifically designed for function-based protein design problems, as opposed to unconditional protein generation.
>
> - First, ProDVa leverages a Text Language Model in three key ways: **(1) to interpret the input textual description, (2) to represent protein fragment descriptions during training, and (3) to retrieve relevant proteins based on functional descriptions during inference.** In the context of unconditional protein generation, none of these functionalities are applicable.
> - Second, the backbone Protein Language Model (ProtGPT2) employed in ProDVa is specifically designed for unconditional generation. Therefore, **ProDVa inherently possesses the capability for unconditional protein generation.**
>
> Despite the above clarifications, we demonstrate how ProDVa can be adapted for unconditional protein generation. By simply fixing the input instruction to "Design a novel protein sequence," the problem addressed by ProDVa effectively reduces to an unconditional protein generation task. To further ensure that the text input as a no-op, we replace the retrieval method with the random selection of fragments. The table below presents the results on unconditional generation. Note that the language alignment metrics are not considered under unconditional settings.
>
> | Models         |    PPL  $\downarrow$  |   pLDDT $\uparrow$  |  % > 70 $\uparrow$  |   PAE  $\downarrow$  |  % < 10 $\uparrow$  |
> | :------------- | :--------: | :-------: | :-------: | :------: | :-------: |
> | Pinal          |   411.93   |   70.11   |   57.02   |  12.76   |   48.44   |
> | ProteinDT (FT) |   593.06   |   47.79   |   0.02    |  26.56   |   0.00    |
> | PAAG (FT)      |  1327.98   |   50.32   |   23.83   |  19.95   |   22.24   |
> | **ProDVa**     | **476.02** | **77.52** | **79.78** | **9.32** | **60.25** |
>
> **ProDVa outperforms all baseline models on the unconditional protein generation task.** Compared to the state-of-the-art baseline, Pinal, ProDVa generates 22.76% and 11.81% more well-folded proteins in terms of pLDDT and PAE, respectively. Compared to other fine-tuned models, ProDVa outperforms them by a large margin. This demonstrates the effectiveness of our method in both conditional and unconditional protein generation settings.
>
> ---
>
> Q3: Explanations of applying ProDVa to Diffusion-based Language Models.
>
> A3: Thanks for your question. ProDVa is specifically designed for autoregressive Transformer architectures. It employs a decoder-only backbone (see Lines 46 and 77) and is trained using curated learning objectives rather than simple next-token prediction (see Lines 92–102). Applying ProDVa to recent Diffusion-based language models remains a non-trivial and challenging task. Nonetheless, we believe this work establishes a foundation for future research on integrating ProDVa with diffusion models.
>
> ---
>
> Q4: Explanations of the generalization under low-resource or novel protein design scenarios where related fragment annotations are unavailable or sparse.
>
> A4: Thanks for your question. In fact, during inference, ProDVa does not rely on any fragment annotations (see Lines 121–123). The training paradigm involves learning functional annotations that are not required during inference. ProDVa learns to select the appropriate token or fragment at each step of decoding. Therefore, under low-resource settings, there is minimal impact on ProDVa's training paradigm. During inference, if no related fragments are provided, ProDVa generates tokens to design a novel protein sequence, similar to ProtGPT2.

---

> > ### Comment · Reviewer_Vivg · 2025-08-04
> >
> > Thanks for your rebuttal and resolving some of my concerns. I have raised my score to 5.

---

### Official Review · Reviewer_VmDi · 2025-07-01

**Clarity:** 3
**Significance:** 3
**Originality:** 3
**Rating:** 5
**Confidence:** 4

**Summary:**

This paper proposes a method to improve the plausibility of protein sequences generated by a protein language model by expanding the vocabulary from the 20 standard amino acids to fragments composed of multiple adjacent amino acids. The approach is designed to address the problem of generating protein sequences conditioned on textual descriptions.

**Questions:**

Q1. Why is the amount of training data used in this work smaller than that used in Pinal?

Q2. How does the choice of different tokenization methods affect the results?

Q3. Does using fragments to represent proteins impact the diversity of the generated sequences?

**Ethical Concerns:**

["NO or VERY MINOR ethics concerns only"]

**Final Justification:**

Their responses addressed most of my concerns. I'll keep my initial positive score.

**Limitations:**

yes

**Quality:**

3

**Strengths And Weaknesses:**

**Strengths**

S1. The algorithm is clearly described and reasonably designed.

S2. The experimental metrics are well constructed and support the conclusions of the paper.

S3. Introducing protein fragments into the generation vocabulary is an interesting and novel attempt, offering new ideas for future research.

**Weaknesses**

W1. The paper mentions that the baseline method Pinal, which achieves comparable performance, uses more training data. However, it does not explain why the training data used in this work is smaller than that of Pinal, nor does it analyze whether increasing the training data might lead to better results than Pinal.

W2. The paper adopts existing tokenization methods but lacks a discussion on whether the choice of tokenizer affects the results.

---

> ### Author Rebuttal · Authors · 2025-07-31
>
> Dear Reviewer VmDi,
>
> First and foremost, we extend our gratitude for your thorough review. Here are our responses to the key points you've raised:
>
> Q1: Explanations of using significantly less training data compared to Pinal.
>
> A1: Thanks for your questions. For the first question:
>
> > Why is the training data used for ProDVa smaller than that used for Pinal?
>
> During training, ProDVa utilizes approximately 712K text-protein pairs, whereas Pinal employs 1.76B pairs. **Despite relying solely on annotated datasets (e.g., SwissProt), Pinal achieves this larger dataset through various data augmentation methods, including the generation (automatic annotation) of textual descriptions for proteins.** For example, it uses models such as GPT-4 and GLM-4-Flash to convert records from SwissProt into complete sentences and domain-specific question-answer pairs that serve as descriptive annotations for different proteins. Additionally, Pinal leverages ProTrek to annotate proteins from UniProtKB and UniRef50, resulting in a large-scale synthetic dataset. **In contrast, our approach primarily focuses on SwissProt and its corresponding annotations, which are manually curated and reviewed by experts.** Our experiments yield the following observations:
>
> - High-quality data (reviewed and manually annotated) is sufficient to train ProDVa to achieve performance comparable to Pinal, which is trained on a large-scale synthetic dataset (model-predicted, unreviewed annotations).
> - ProDVa demonstrates data efficiency, utilizing only 0.02%–0.04% of the data used by Pinal, as it does not rely on synthetic annotations.
> - The dataset used for ProDVa is entirely open-source, and its acquisition is straightforward and time-efficient, in contrast to Pinal, which depends on LLMs or other oracle models for data generation.
>
> For the second question:
>
> > Whether increasing the training data might lead to better results than Pinal?
>
> We have discussed the question in our manuscript (Lines 229-241). Here is a summary:
>
> - Unfortunately, the large-scale datasets used by Pinal have not yet been open-sourced. Reproducing the synthetic annotations described above is challenging for us, as the process is both costly (e.g., requiring the use of LLM APIs and deployment of oracle models) and time-consuming (e.g., UniRef50 alone has over 41M proteins, yet UniProtKB includes more than 173M).
> - In Table 2, we demonstrate that ProteinDT and PAAG perform significantly better after fine-tuning with additional data (the same dataset used by ProDVa), particularly with respect to Language Alignment metrics. Therefore, we carefully use the phrase "may potentially" to conclude that incorporating more data may improve performance.
>
> ---
>
> Q2: Clarifications on choosing the tokenizer (tokenization method).
>
> A2: Thanks for your question. Tokenization in our work involves two aspects: tokenizing amino acids and tokenizing fragments.
>
> First, for amino acid tokenization, since ProDVa utilizes ProtGPT2 as the PLM backbone, we employ the ProtGPT2 tokenizer (a BPE tokenizer) for both tokenizing amino acids and constructing the static protein vocabulary. Alternative tokenization methods (e.g., per-amino-acid tokenization) are not suitable for our purposes.
>
> Second, for fragment tokenization, we introduce our methods in Section 2, based on the intuition and empirical study presented in Section 1 (see Figure 1). Additional experiments in Appendix E.5 demonstrate that our tokenization method exhibits a degree of robustness to noise when encountering irrelevant fragments or random N-gram subsequences.
>
> ---
>
> Q3: Evaluations on Sequence Diversity.
>
> A3: Thanks for your valuable feedback. Using fragments to represent proteins has minimal impact on the diversity of designed sequences. In contrast, our training paradigm enables ProDVa to design user-specified proteins while maintaining diversity in the generated sequences.
>
> We have included experiments and analyses on Sequence Diversity **in our response to Reviewer Sz6U in Q1**. Please refer to that response for more details. A summary of the key observations is provided below:
>
> - ProDVa exhibits comparable sequence diversity to all other baseline models. This demonstrates that ProDVa is capable of designing diverse sequences that satisfy the same input descriptions.
> - Under the same training settings, ProDVa maintains high sequence diversity, outperforming these two fine-tuned baselines by 14.14% and 7.92%, respectively. This underscores ProDVa’s ability to design protein sequences that satisfy user-specified requirements while preserving diversity in the generated sequences.

---

> > ### Comment · Reviewer_VmDi · 2025-08-04
> > **Response to the authors**
> >
> > Thanks for your detailed rebuttal, which addressed most of my concerns. I'll keep my score.

---

### Official Review · Reviewer_Dwmc · 2025-07-01

**Clarity:** 3
**Significance:** 3
**Originality:** 3
**Rating:** 5
**Confidence:** 4

**Summary:**

The paper proposes a novel approach for function-based protein design, named PRODVA, that integrates (1) a text encoder, (2) a protein language model (PLM), and (3) a fragment encoder. Unlike prior methods, PRODVA dynamically incorporates fragments of natural proteins—retrieved based on functional annotations—to enhance structural plausibility and language alignment. The model demonstrates strong performance across multiple benchmarks (CAMEO and Mol-Instructions), significantly outperforming previous approaches (e.g., ProteinDT, PAAG, and even Pinal) in terms of foldability (pLDDT, PAE) while using a fraction of the training data.

**Questions:**

Have the authors considered weighting fragments differently based on type (e.g., active site vs. repeat)? Does one category contribute more to structural accuracy?

**Ethical Concerns:**

["NO or VERY MINOR ethics concerns only"]

**Limitations:**

yes

**Paper Formatting Concerns:**

Some figure captions are insufficiently descriptive. Consider expanding them (e.g., Figures 1b/c).

**Quality:**

4

**Strengths And Weaknesses:**

Strengths
1.	The integration of dynamic fragment vocabulary based on functional annotations is both novel and biologically grounded.
2.	 PRODVA achieves superior foldability and competitive functional alignment, outperforming state-of-the-art models despite using substantially less training data.
Weaknesses
1.	Some figure captions are insufficiently descriptive. Consider expanding them (e.g., Figures 1b/c).
2.	It is recommended to evaluate the model on out-of-distribution settings, such as unseen protein families or novel synthetic functions.

---

> ### Author Rebuttal · Authors · 2025-07-31
>
> Dear Reviewer Dwmc,
>
> First and foremost, we extend our gratitude for your thorough review. Here are our responses to the key points you've raised:
>
> Q1: Expanding the figure captions to provide more descriptive information.
>
> A1: Thanks for your valuable feedback. We will expand the figure captions in the final version as follows:
>
> - Figure 1(b): Performance on pLDDT (higher is better). Our method significantly improves the pLDDT of proteins generated by Random+ by a large margin of 12%, with the average pLDDT exceeding the well-folded threshold.
> - Figure 1(c): Performance on PAE (lower is better). Compared to Random+, our method achieves a 9% improvement in PAE and is the only model to surpass the well-folded threshold. Notably, our method outperforms the state-of-the-art baseline model Pinal in both pLDDT and PAE.
>
> ---
>
> Q2: Evaluations on designing proteins under out-of-distribution settings (e.g., evaluating on novel synthetic functions).
>
> A2: Thanks for your valuable feedback. To evaluate performance in designing novel proteins under out-of-distribution (OOD) settings, we first construct a test set comprising 1,000 randomly synthesized novel functions. The Mol-Instructions test set is utilized, and all unique functions are extracted. For each synthetic function, 2–5 different functions are randomly selected and concatenated into a single composite function. Due to the random selection, combination, and arrangement of functions, the resulting synthetic functions are guaranteed to be novel and under OOD settings.
>
> Some key findings can be summarized as follows:
>
> - **ProDVa outperforms other baseline models under OOD settings, including the state-of-the-art baseline Pinal, across both foldability and language alignment metrics.**
> - In terms of foldability, both ProDVa and Pinal demonstrate promising performance, indicating that OOD settings do not affect a model’s ability to design structurally plausible proteins.
> - For language alignment, all models perform worse than on the Mol-Instructions test set (see Section). This may be attributed to two main factors. First, the OOD settings pose significant challenges for current models. Understanding synthetic functions that may not exist in the real world likely requires further investigation in future work. Second, the OOD settings may not be properly validated, as verifying the plausibility of these synthetic functions is inherently difficult (e.g., may require further wet-lab experiments). Moreover, natural proteins with such properties may not exist, potentially rendering the design of proteins with novel synthetic functions unfeasible.
>
> | Models         |    PPL $\downarrow$     |   pLDDT $\uparrow$  |   PAE $\downarrow$    | ProTrek Score $\uparrow$ | EvoLlama Score $\uparrow$ |
> | :------------- | :--------: | :-------: | :------: | :-----------: | :------------: |
> | Pinal          |   326.79   |   74.75   |  11.29   |     9.23      |     48.47      |
> | ProteinDT (FT) |  1877.44   |   34.38   |   24.3   |     4.03      |     43.06      |
> | PAAG (FT)      |  1406.22   |   48.65   |  20.88   |     4.19      |     45.89      |
> | **ProDVa**     | **412.27** | **77.41** | **9.11** |   **9.79**    |   **49.17**    |
>
> ---
>
> Q3: Explanations of weighting fragment types.
>
> A3: Thanks for your question. As mentioned in our manuscript (Lines 105–110), the fragment types in our training set (CAMEO subset and Mol-Instructions) are inherently unbalanced. Enforcing a balanced distribution of fragment types would require removing up to 80% of the training data, potentially resulting in insufficient data for training. To address this issue, we introduce a weighted type loss $\mathcal{L}_{\text{TYPE}}$, as described in Equation (4) (see Lines 105–110).

---

> > ### Comment · Reviewer_Dwmc · 2025-08-05
> >
> > Thanks for your detailed rebuttal, which addressed most of my concerns. I'll keep my score.

---

### Official Review · Reviewer_Q2yf · 2025-07-03

**Clarity:** 3
**Significance:** 3
**Originality:** 3
**Rating:** 4
**Confidence:** 5

**Summary:**

The authors propose PRODVA, a text-guided protein design model that has a fragment encoder to retrieve protein fragments based on textual descriptions. The advantages of PRODVA is it achieves similar functional alignment with 0.04% training data, and achieving high pLDDT and PAE.

**Questions:**

See Weaknesses.

**Ethical Concerns:**

["NO or VERY MINOR ethics concerns only"]

**Final Justification:**

I do not have any other concern now.

**Limitations:**

yes

**Paper Formatting Concerns:**

None.

**Quality:**

2

**Strengths And Weaknesses:**

Strengths:

1. The PRODVA achieves high foldability and comparable 'language alignment' with limited size of training set.

2. The presentation is easy to understand.

3. The paper introduces fragments into protein design, which is technically sound and interesting.

Weaknesses:

1. Since PRODVA designs proteins by retrieval from existing fragments, e.g., motifs and functional sites, one main concern is, it seems PRODVA cannot designs proteins with novel fragments. It raises the concern that the proteins generated by PRODVA may has not satisfactory novelty.

2. In Table 1, because the metrics are evaluated by other deep learning model, these metrics are not the higher/lower the better. These metrics need to be close to the distribution of natural proteins.

3. As PRODVA is initilized by ProtGPT2, I am not sure if that's appropriate to also employ ProtGPT2 to test the PPL. Could the authors have more discussions on this point?

4. The work defines 7 fragment types, including family. Does it more like a property of protein instead of a fragment?

5. In 'Language Alignment' metrics, PRODVA excels at retrieval accuracy. Is this because PRODVA itself is retrieving fragments from existing protein dataset, and these fragments are easier to be identified by protein encoder than the proteins generation from scratch?

---

> ### Author Rebuttal · Authors · 2025-07-31
>
> Dear Reviewer Q2yf,
>
> First and foremost, we extend our gratitude for your thorough review. Here are our responses to the key points you've raised:
>
> Q1: Explanations of designing proteins with novel fragments.
>
> A1: Thanks for your question. The novel protein sequences designed by ProDVa comprise two components: sets of tokens (amino acids) and sets of fragments. The novelty of these sequences is determined by both factors. In terms of tokens, ProDVa possesses capabilities comparable to those of ProtGPT2. In terms of fragments, fragment candidates are dynamically retrieved from a database (supporting documents) during inference. **In our experimental settings, the retrieval database contains over 1 million unique fragments (1.16M for the function keyword setting and 1.43M for the textual description setting). Consequently, ProDVa is capable of designing novel protein sequences.**
>
> Additionally, we will explain our choice of using a Fragment Encoder instead of a Fragment Decoder, which might otherwise generate novel fragments. We randomly mutate the amino acids in fragments of the protein sequences designed by ProDVa with a 40% probability, replacing each amino acid with a different one. The table below presents a comparison between ProDVa and simple mutation on fragments. **Both foldability and language alignment perform significantly worse under random mutation. These results highlight the challenges of incorporating novel fragments and indicate that generating novel fragments may not be suitable for designing user-specified protein sequences.**
>
> | Models       |  PPL $\downarrow$   | pLDDT $\uparrow$ |  PAE $\downarrow$  | ProTrek Score $\uparrow$ | EvoLlama Score $\uparrow$ |
> | :----------- | :----: | :---: | :---: | :-----------: | :------------: |
> | ProDVa       | 415.63 | 76.86 | 8.66  |     17.40     |     51.10      |
> | ProDVa (Mut) | 4168.9 | 44.34 | 17.55 |     7.71      |     41.03      |
>
> ---
>
> Q2: Clarifications on how to interpret the results and metrics in Table 1.
>
> A2: Thanks for your valuable feedback. The metrics presented in Tables 1 and 2 are used to evaluate the quality of the designed proteins and to compare them with various baseline models. Admittedly, interpreting these metrics solely based on whether higher or lower values are better may not always be appropriate. To more intuitively illustrate performance differences among models, we report the absolute values of each metric rather than their relative differences (i.e., delta values) from natural proteins. This approach is consistent with prior studies (e.g., ProteinDT, Pinal), which also present absolute scores. Due to space constraints and to maintain the simplicity of our table presentations, delta values were not included in the current version. We will consider incorporating them in the final version. In this way, we believe that comparisons with other models using absolute values, as well as comparisons with natural protein distributions using delta values, are both intuitive.
>
> ---
>
> Q3: Discussions and evaluations on computing the PPL metric with the alternative Protein Language Model.
>
> A3: Thanks for your valuable feedback. To evaluate the impact of different Protein Language Models on the computation of the PPL metric, we selected three additional models: ProGen2 [1], RITA [2], and ProteinGLM [3]. The table below presents supplementary results for the PPL metric (i.e., PPL (ProtGPT2)) as referenced in Table 2.
>
> | Models         | PPL (ProtGPT2) $\downarrow$  | PPL (ProGen2) $\downarrow$ | PPL (RITA) $\downarrow$  | PPL (ProteinGLM) $\downarrow$  |
> | :------------- | :------------: | :-----------: | :--------: | :--------------: |
> | Natural        |     318.15     |     5.99      |    5.52    |      16.86       |
> | Random (U)     |    2484.03     |     21.71     |   22.14    |      40.87       |
> | Random (E)     |    3136.88     |     18.68     |   19.04    |      32.95       |
> | Random+ (E)    |     846.01     |     10.08     |    9.32    |       8.77       |
> | ProteinDT      |    1576.23     |     12.41     |   12.44    |      18.41       |
> | ProteinDT (FT) |    1213.38     |     10.80     |   10.69    |      19.98       |
> | Pinal          |     308.97     |     5.81      |    5.77    |      17.77       |
> | PAAG           |    2782.70     |     17.84     |   18.05    |      24.44       |
> | PAAG (FT)      |    1332.35     |     11.09     |   11.03    |      19.28       |
> | Chroma         |    1370.21     |     12.22     |   12.42    |      38.59       |
> | **ProDVa**     |   **415.63**   |   **7.63**    |  **8.82**  |    **31.39**     |
>
> **The results are consistent with the PPL values computed by ProtGPT2, despite differences in absolute scores. ProDVa consistently remains within a low PPL range and is closer to natural proteins.**
>
> ---
>
> Q4: Explanations of fragment types.
>
> A4: Thanks for your question. In our manuscript, we define fragments of a protein as continuous subsequences (see Lines 58–59), implying that any arbitrary subsequence can be considered a fragment. However, fragments lacking functional annotations are not significant for designing proteins with user-specified requirements. Therefore, identifying usable fragments is a central challenge in our work. In Section 2 and Appendix B, we describe how InterPro is employed to extract fragments with functional annotations. Moreover, a single protein may contain multiple fragments, each potentially associated with different properties (or functions). Thus, we consider "fragment" to be the most appropriate term, as the fragments used in our work possess diverse functional annotations.
>
> ---
>
> Q5: Further discussions on the results of Retrieval Accuracy metric.
>
> A5: Thanks for your question. We argue that ProDVa’s high Retrieval Accuracy is unrelated to retrieving fragments from existing protein datasets. The metric used (see Lines 525–534 for details) evaluates whether the designed protein shows the highest similarity to the textual description among a set of randomly selected candidates. Similarity is computed using the ProTrek model, which is not involved in the training or inference of ProDVa, thereby avoiding evaluation bias. Moreover, whether the sequences are designed from scratch or generated by ProDVa, they more or less exhibit features of natural proteins (e.g., evolutionary information). Importantly, the candidate pool for Retrieval Accuracy consists entirely of natural proteins, ensuring that the metric provides an unbiased comparison of protein sequences generated by different models.
>
> References:
>
> [1] Nijkamp, Erik, et al. "Progen2: exploring the boundaries of protein language models." Cell systems 14.11 (2023): 968-978.
>
> [2] Hesslow, Daniel, et al. "Rita: a study on scaling up generative protein sequence models." arXiv preprint arXiv:2205.05789(2022).
>
> [3] Chen, Bo, et al. "xTrimoPGLM: unified 100B-scale pre-trained transformer for deciphering the language of protein." arXiv preprint arXiv:2401.06199 (2024).

---

> > ### Comment · Reviewer_Q2yf · 2025-08-01
> > **Response**
> >
> > Thanks for the authors' rebuttal. My concerns have been addressed, and I will increase the rate to 4.

---

### Official Review · Reviewer_Sz6U · 2025-07-10

**Clarity:** 3
**Significance:** 2
**Originality:** 2
**Rating:** 5
**Confidence:** 2

**Summary:**

The manuscript focuses on protein design, specifically on generating sequences with desirable properties specified through text inputs. With common methods, such sequences often suffer from poor in silico folding metrics. The authors address this issue by incorporating protein fragments (e.g., motifs, functional sites), which helps improve folding quality.

**Questions:**

- The embeddings of random protein sequences using ESM-C might be misleading, or at least should be interpreted cautiously. ESM-C may not have encountered such proteins during training and could project them into similar embedding regions. Using embeddings from Pinal might provide a more accurate comparison and a better sense of the embedding space's range.

- In Figure 1a, the orange color is used for both random and very low pLDDT sequences. I understand that one corresponds to structures and the other to models, but using distinct colors would help avoid confusion.

- In Equations (1) and (2), is something missing on the left side of the curly bracket?

- On line 86, are the representations mapped to the same embedding space as the protein language model via an adapter, or do they already share the same dimensional space?

**Ethical Concerns:**

["NO or VERY MINOR ethics concerns only"]

**Final Justification:**

My main concern was around the evaluation, especially sequence diversity paired with other metrics. The authors provided additional quantitative metrics and addressed my questions.

**Limitations:**

The limitations of the proposed method should be discussed explicitly in the conclusion or highlighted through specific experiments.

**Paper Formatting Concerns:**

No concerns.

**Quality:**

3

**Strengths And Weaknesses:**

The manuscript is overall easy to follow, and the method is clearly described. The proposed approach appears to outperform existing baselines across different tasks.

Weaknesses

Sequence diversity should be reported alongside other metrics. Additionally, It would be interesting to report pAE (or other structural metrics) in terms of clusters of sequences.

I may have missed this, but is the proposed method restricted to single-chain proteins? If not, evaluating interaction predictions (e.g., by reporting pae_interaction on known interacting chains) would strengthen the work.

---

> ### Author Rebuttal · Authors · 2025-07-31
>
> Dear Reviewer Sz6U,
>
> First and foremost, we extend our gratitude for your thorough review. Here are our responses to the key points you've raised:
>
> Q1: Evaluations on Sequence Diversity and structure-based metrics in terms of clutsters of sequences.
>
> A1: Thanks for your valuable feedback.
>
> > Sequence diversity should be reported alongside other metrics.
>
> We first present the experimental results on Sequence Diversity. MMseqs2 [1] is used to compute the similarity $\mathrm{sim}$ between each pair of sequences in a batch of size $N$, where all sequences in the batch are designed based on the same textual description (i.e., with different random seeds). Sequence Diversity is formalized as follows:
>
> $$
> \text{Sequence Diversity} = \frac{\sum_i (1 - \mathrm{sim}_i)}{N(N - 1)}
> $$
>
> The results (in %) are presented in the table below. Key observations are summarized as follows:
> - **ProDVa exhibits comparable sequence diversity to all other baseline models.** Compared to the state-of-the-art baseline model Pinal, our method achieves similar performance on the Mol-Instructions dataset and outperforms it by over 15% on the CAMEO subset. **This demonstrates that ProDVa is capable of designing diverse sequences that satisfy the same input descriptions.**
> - On the Mol-Instructions dataset, we observe that both ProteinDT and PAAG exhibit significantly reduced sequence diversity after fine-tuning, with decreases of 19.36% and 13.06%, respectively. **In contrast, under the same training settings, ProDVa maintains high sequence diversity, outperforming these two fine-tuned baselines by 14.14% and 7.92%, respectively. This underscores ProDVa’s ability to design protein sequences that satisfy user-specified requirements while preserving diversity in the generated sequences.**
>
> | Models         | Diversity $\uparrow$ (CAMEO Subset) | Diversity $\uparrow$ (Mol-Instructions) |
> | -------------- | :----------------------: | :--------------------------: |
> | Random (U)     |          97.46           |            97.01             |
> | Random (E)     |          99.78           |            99.56             |
> | Random+ (E)    |          98.85           |            98.63             |
> | ProteinDT      |          99.72           |            99.23             |
> | ProteinDT (FT) |          99.32           |            79.87             |
> | Pinal          |          82.72           |            95.28             |
> | PAAG           |          99.02           |            99.15             |
> | PAAG (FT)      |          99.87           |            86.09             |
> | Chroma         |          98.12           |            97.00             |
> | ESM3           |          96.75           |              -               |
> | **ProDVa**     |        **98.58**         |          **94.01**           |
>
> > Additionally, It would be interesting to report pAE (or other structural metrics) in terms of clusters of sequences.
>
> To evaluate foldability in terms of clusters of sequences, we first use MMSeqs2 to cluster sequences at various identity thresholds (indicated in parentheses). The average pLDDT across clusters is defined as follows (similarly for PAE across clusters):
>
> $$
> \text{pLDDT across clusters} = \frac{1}{k} \sum_{j=1}^{k} \left( \frac{1}{n_j} \sum_{i=1}^{n_j} \text{pLDDT}_{ij} \right)
> $$
>
> where $k$ is the number of clusters, and $n_j$ denotes the number of proteins in the $j$-th cluster.
>
> | Models         |   pLDDT $\uparrow$   | pLDDT (30%) $\uparrow$ | pLDDT (50%) $\uparrow$ |   PAE $\downarrow$   | PAE (30%) $\downarrow$ | PAE (50%) $\downarrow$ |
> | -------------- | :-------: | :---------: | :---------: | :------: | :-------: | :-------: |
> | Natural        |   80.64   |    77.24    |    80.44    |   9.20   |   11.25   |   9.61    |
> | Random (U)     |   22.96   |    22.96    |    22.96    |  24.85   |   24.85   |   24.85   |
> | Random (E)     |   25.77   |    25.77    |    25.77    |  24.71   |   24.71   |   24.71   |
> | Random+ (E)    |   64.47   |    64.32    |    64.40    |  17.91   |   18.01   |   17.96   |
> | ProteinDT      |   38.29   |    38.10    |    38.16    |  25.13   |   25.19   |   25.19   |
> | ProteinDT (FT) |   51.42   |    40.53    |    43.57    |  18.57   |   22.50   |   21.41   |
> | Pinal          |   75.25   |    66.90    |    67.87    |  10.96   |   14.81   |   14.41   |
> | PAAG           |   28.39   |    28.39    |    28.39    |  25.38   |   25.38   |   25.38   |
> | PAAG (FT)      |   50.37   |    39.93    |    43.17    |  19.96   |   24.14   |   22.85   |
> | Chroma         |   59.18   |    59.13    |    59.17    |  15.03   |   15.06   |   15.03   |
> | **ProDVa**     | **76.86** |  **74.51**  |  **76.68**  | **8.66** | **9.09**  | **8.74**  |
>
> **The results indicate that ProDVa consistently outperforms all baseline models in both overall foldability and foldability across clusters.** At lower identity thresholds, the pLDDT decreases slightly compared to fine-tuned ProteinDT and PAAG, demonstrating ProDVa's robustness across diverse designed protein sequences.
>
> We will include the results above in the final version.
>
> ---
>
> Q2: Clarifications on the applicability of ProDVa.
>
> A2: Thanks for your question. ProDVa is proposed for the design of single-chain proteins, consistent with previous work such as ProteinDT, PAAG, and Pinal.
>
> ---
>
> Q3: Explanations of using ESM C as the embedding model in Figure 1.
>
> A3: Thanks for your question. There are two main reasons why we prefer to use ESM C instead of Pinal.
>
> **First, Pinal is a text-to-protein model that takes natural language as input and is not specifically designed to compute protein embeddings.** It comprises two main modules: a text-to-structure module and a SaProt [2] module, of which only the latter can be used as an embedding model.
>
> **Second, technically speaking, computing protein embeddings using ESM C and Pinal (the SaProt module) exhibits very similar behavior, including for those generated by Random(+) and ProDVa.** This similarity arises because Pinal was trained on proteins from SwissProt, UniRef50, and UniProtKB. Consequently, both ESM C and Pinal are unlikely to have encountered the remaining proteins, whether random or novel.
>
> Therefore, we believe that ESM C is a more suitable choice for embedding the landscape of random proteins and novel proteins designed by ProDVa, as it can offer more meaningful insights for comparison with natural proteins.
>
> **To validate this, we use Pinal (the SaProt module) as the embedding model to visualize the landscape, following the approach used in Figure 1(a). The visualization results are highly consistent with those obtained using ESM C. Proteins generated by Random are clustered, while those from Random+ exhibit a significantly more diverse distribution. As expected, proteins designed by ProDVa span the landscape of natural proteins. The visualization results using Pinal will be updated in the final version.**
>
> ---
>
> Q4: Clarifications on the colors chosen for legends used in Figure 1(a).
>
> A4: Thanks for your valuable feedback. The legend colors in the upper right are consistent with AlphaFold's color scheme. We will adjust the legend color scheme for scatter points in the final version.
>
> ---
>
> Q5: Clarifications on the Equations (1) and (2).
>
> A5: Sorry for misunderstanding. There is nothing missing on the left side of the curly brackets in Equations (1) and (2). They denote a group of variables.
>
> ---
>
> Q6: Further explanations of the representations on Line 86.
>
> A6: Sorry for misunderstanding. The projection layer is optional. If the Fragment Encoder shares the same dimension as the PLM (i.e., $d_{\text{PLM}}$), the use of the adapter is determined by the user. Otherwise, the projection layer is employed to map the fragment embedding matrix (representing $m$ fragments) to the same embedding space as the PLM.
>
> References:
>
> [1] Steinegger, Martin, and Johannes Söding. "MMseqs2 enables sensitive protein sequence searching for the analysis of massive data sets." Nature biotechnology 35.11 (2017): 1026-1028.
>
> [2] Su, Jin, et al. "Saprot: Protein language modeling with structure-aware vocabulary." BioRxiv (2023): 2023-10.

---

> > ### Comment · Reviewer_Sz6U · 2025-08-04
> >
> > Thank you for addressing my comments and questions.
> >
> > About the clustering, I think it is more informative to cluster *after* ranking with a score. For example, clustering all the sequences with a pAE score lower than 10. Otherwise, the metric can still be dominated by a few "good" clusters.
> >
> > I don't have additional questions at the moment, but I will keep following the discussion.

---

> > > ### Author Response · Authors · 2025-08-04
> > >
> > > Dear Reviewer Sz6U,
> > >
> > > Thank you for your time and effort in engaging with us during the discussion period. Below is our responses to your comments regarding the clustering.
> > >
> > > We compute the average pLDDT and PAE after clustering the proteins based on their sequence identity (see Q1 in the rebuttal) because most baseline models struggle to design well-folded proteins (i.e., pLDDT > 70 and PAE < 10). As shown in Tables 1 and 2 of our manuscript, except for ProDVa and Pinal, the other baselines rarely design more than 50% well-folded proteins. Therefore, **we perform clustering prior to computing the metrics to enable a fair comparison across all sequences, regardless of whether the proteins are well-folded.**
> > >
> > > To mitigate the potential impact of structural metrics being dominated by a few "good" or "bad" clusters, we first filter out well-folded and not well-folded proteins, and then compute the average pLDDT and PAE across the filtered clusters. Note that
> > >
> > > - **The filtering operation is based on the well-folded threshold (70 for pLDDT and 10 for PAE).** Filtering is conducted separately, resulting in four distinct filtered clusters.
> > > - **The results reflect the structural plausibility of designed proteins within each filtered cluster. For evaluating model performance, we consider the average scores across all sequences (without filtering) to be a better choice.**
> > >
> > > | Models         | pLDDT\* (30%) $\uparrow$ | pLDDT\* (50%) $\uparrow$ | PAE\* (30%) $\downarrow$ | PAE\* (50%) $\downarrow$ | pLDDT\*\* (30%) $\uparrow$ | pLDDT\*\* (50%) $\uparrow$ | PAE\*\* (30%) $\downarrow$ | PAE\*\* (50%) $\downarrow$ |
> > > | -------------- | :----------: | :----------: | :--------: | :--------: | :-----------: | :-----------: | :---------: | :---------: |
> > > | Natural        |    87.02     |    88.54     |    4.93    |    4.52    |     67.04     |     67.79     |    17.83    |    17.59    |
> > > | Random (U)     |    74.31     |    74.31     |    5.34    |    5.34    |     22.87     |     22.87     |    24.88    |    24.88    |
> > > | Random (E)     |    76.77     |    76.77     |    4.60    |    4.60    |     25.68     |     25.68     |    24.74    |    24.74    |
> > > | Random+ (E)    |    81.59     |    81.93     |    7.02    |    6.89    |     63.19     |     63.20     |    18.73    |    18.72    |
> > > | ProteinDT      |    80.32     |    80.36     |    6.17    |    6.13    |     38.04     |     38.10     |    25.22    |    25.21    |
> > > | ProteinDT (FT) |    81.75     |    84.51     |    5.95    |    5.65    |     39.47     |     40.17     |    22.93    |    22.73    |
> > > | Pinal          |    85.27     |    85.56     |    5.40    |    5.29    |     57.71     |     57.59     |    19.57    |    19.73    |
> > > | PAAG           |    76.09     |    76.09     |    6.15    |    6.15    |     28.37     |     28.37     |    25.39    |    25.39    |
> > > | PAAG (FT)      |    83.15     |    84.89     |    5.67    |    5.55    |     39.14     |     39.87     |    24.48    |    24.23    |
> > > | Chroma         |    75.89     |    75.89     |    5.81    |    5.81    |     55.36     |     55.37     |    17.13    |    17.13    |
> > > | **ProDVa**     |  **83.20**   |  **84.36**   |  **4.72**  |  **4.56**  |   **64.59**   |   **66.49**   |  **14.19**  |  **14.37**  |
> > >
> > > - \* denotes the filtered clusters are well-folded (pLDDT > 70 or PAE < 10)
> > > - \*\* denotes the filtered clusters are not well-folded (pLDDT < 70 or PAE > 10)
> > >
> > > We highlight the key findings as follows:
> > >
> > > - For well-folded proteins (indicated by \*), there is no significant gap across different models. However, it is important to note that for some baselines (e.g., Random, ProteinDT, and PAAG), the proportion of well-folded proteins is significantly low (see Table 2 in our manuscript, less than 1%).
> > > - **For proteins that are not well-folded (indicated by \*\*), ProDVa outperforms other models by a substantial margin, with average scores across clusters approaching the threshold. This highlights the robustness of our method in designing structurally plausible proteins.**

---

> > > > ### Author Response · Authors · 2025-08-06
> > > >
> > > > Dear Reviewer Sz6U,
> > > >
> > > > Thank you once again for your time and effort. As the discussion period comes to an end, we would appreciate it if you could let us know whether our responses above have adequately addressed the points you raised, or if there are any additional questions or feedback we should address.

---

### Decision · Program_Chairs · 2025-09-17

**Decision:**

Accept (spotlight)

**Comment:**

This paper presents ProDVa, a protein design framework that integrates text-based functional descriptions with fragment information from natural proteins to improve structural plausibility. By combining a text encoder, a protein language model, and a fragment encoder, the method achieves comparable function alignment to state-of-the-art models while using less than 0.04% of the data, and significantly improves foldability (e.g., +7.38% pLDDT > 70).
During the review process, all reviewers gave a positive score to this paper, recognizing its novelty, technical contribution, performance, and writing. From my perspective, integrating functional fragments through a dynamic vocabulary is novel. It may greatly improve foldability and functional alignment. The comprehensive experiments validate this point. Therefore, I recommend the acceptance of this paper for a spotlight presentation.